

# Passive Microwave Remote Sensing based High Resolution Snow Depth Mapping for Western Himalayan Zones using Multifactor Modelling Approach

Dhiraj Kumar Singh[1], Srinivasarao Tanniru[1], Kamal Kant Singh[3], Harendra Singh Negi[3], RAAJ Ramsankaran[1,2*]

[1]Hydro-Remote Sensing Applications(H-RSA) Group, Department of Civil Engineering, Indian Institute of Technology Bombay, Powai, Mumbai 400076, India.
[2] Interdisciplinary Program in Climate Studies, Indian Institute of Technology Bombay, Powai, Mumbai 400076, India.
[3]Defence Geoinformatics Research Establishment, Him Parisar, Sector 37A, Chandigarh 160036, India

*Correspondence to*: RAAJ Ramsankaran (ramsankaran@civil.iitb.ac.in)

**Abstract.** Spatiotemporal snow depth (SD) mapping in the Indian Western Himalayan (WH) region is essential in many applications pertaining to hydrology, natural disaster management, climate, etc. In-situ techniques for SD measurement are not sufficient to represent the high spatiotemporal variability of SD in WH. Currently, low-frequency passive microwave (PMW) remote sensing-based algorithms are extensively used to monitor SD at regional and global scales. However, only a limited number of PMW SD estimation studies are carried out for WH till date. In addition, the majority of the available PMW SD models for WH locations are developed using limited data and less parameters, therefore cannot be implemented for the entire region. Further, these models have not considered the auxiliary parameters such as location, topography, snow cover days (SCD) into consideration and have poor accuracy (particularly in deep snow), and coarse spatial resolution.

Considering the high spatiotemporal variability of snow depth characteristics across WH region, region wise multifactor models are developed for the first time to estimate SD at high spatial resolution of 500 m x 500 m for three different WH zones i.e., Lower Himalayan Zone (LHZ), Middle Himalayan Zone (MHZ), and Upper Himalayan Zone (UHZ). Multifrequency brightness temperature (TB) observations from Advanced Microwave Scanning Radiometer 2 (AMSR2), SCDs data, terrain parameters (i.e., elevation, slope and ruggedness), geolocation for the winter period (October to March) during 2012-13 to 2016-17 are used for developing the SD models. Different regression approaches (i.e., linear, logarithmic, reciprocal, and power) are developed and evaluated to find if any of these models can address the heterogeneous association between SD observations and PMW TB. The results indicate the following observations: (a) multifactor model developed using power regression has shown improved accuracy in SD retrievals compared to other regression approaches in all WH zones; (b) spatial variability in SD is highly affected by SCDs, terrain parameters, geolocation parameters; (c) compared to the currently operational AMSR2 SD products, the proposed models have shown better SD estimates in all WH zones particularly when SD > 25 cm; (d) the Root Mean Square Error (RMSE) of multifactor models SD estimates increased with an increase in SCD in all WH zones; The multifactor model of MHZ has shown lesser RMSE (i.e., 27.21 cm) compared to



LHZ (32.87 cm) and UHZ (42.81 cm). Overall results indicate that the proposed multifactor SD models have achieved higher accuracy in deep snowpack (i.e., SD >25 cm) of WH when compared to various previously developed SD models.

**Introduction**

Snow is an essential land cover type and an important cryosphere component. The snow cover encompasses an aerial extent of approximately $45 \times 10^6$ km$^2$ in the peak winter over the northern hemisphere (Estilow et al., 2015; Lemke et al., 2007). Among many cryosphere regions in the northern hemisphere, the Indian Western Himalayan (WH) is a unique snow-covered region with a complex topography and high spatiotemporal variability in snow depth (SD), and diverse land cover types (Singh et al., 2018; Das and Sarwade, 2008; Thakur et al., 2019; Sharma et al., 2014; Singh et al., 2016). WH comprises

three mountain zones, e.g., Lower Himalayan Zone (LHZ), Middle Himalayan Zone (MHZ), and Upper Himalayan Zone (UHZ), and receives significant snowfall during winter (Dimri and Dash, 2012; Gurung et al., 2011; Kumar et al., 2019; Sharma and Ganju, 2000; Singh et al., 2016, 2014). The variation in snow volume and its melt rate affects the availability of fresh water for drinking, hydropower, irrigation facilities, and ecosystem conditions for millions of people residing in the foothills of WH zones (Singh et al., 2016; Thakur et al., 2019; Nüsser et al., 2019; Negi et al., 2020; Ahmad, 2020; Mukherji

et al., 2019; Vishwakarma et al., 2022). Further, the variability in snow characteristics such as SD, density, volume, etc., and mountainous topography triggers frequent avalanches in the WH region, which have resulted in more than 1000 casualties as reported in different studies (Ganju et al., 2004; McClung, 2016; Gusain et al., 2016). Therefore, quantifying snow variables, especially SD, is an essential field of study in the WH.

Traditionally SD information is acquired using in-situ measurements from snow stakes, snow poles, ground penetrating

radar, automatic weather stations, etc. (Dong, 2018; Kinar and Pomeroy, 2015). In-situ methods provide accurate SD; however, these techniques have several drawbacks, such as limited spatial coverage, operational and maintenance constraints under harsh weather and complex terrain conditions, instrument calibration and malfunctioning issues, and high logistics and personnel requirements (Kinar and Pomeroy, 2015; Gusain et al., 2016). In WH, because of the rocky terrain and harsh climatic conditions, a sparse network of snow monitoring stations is available (Saraf et al., 1999; Singh et al., 2016; Gusain

et al., 2016). Apart from this, the available SD observations from the in-situ network are spatially and temporally discontinuous and inadequate for demonstrating the snowpack at a regional scale, particularly in the high-altitude regions of WH. Space-borne passive microwave (PMW) remote sensing observations can partially compensate for these limitations and effectively monitor large areas with SD at a comparatively low cost under all weather and terrain conditions (Dietz et al., 2012; Amlien, 2008; Bernier, 1987; Xiao et al., 2018). Sensitivity to snowpack characteristics, global coverage, daily

temporal resolution, and availability of extensive archive of historical data makes space-borne PMW remote sensing data





extensively useful for the retrieval of SD (Dietz et al., 2012; Tedesco and Narvekar, 2010; Luojus et al., 2021; Chang et al., 1987).

The historical PMW data, ongoing and planned missions have paved the way for developing numerous SD inversion algorithms across the different cryosphere regions of the earth. Many studies for SD estimation have been carried out using
multifrequency brightness temperature (TB) observations collected from PMW sensors onboard different satellites (Chang et al., 1987; Saraf et al., 1999; Xiao et al., 2018; Kelly et al., 2005; Takala et al., 2011; Dai et al., 2018; Jiang et al., 2014; Singh et al., 2012). The volumetric PMW scattering increases, whereas PMW TB decreases with an increase in SD. The PMW brightness temperature difference (BTD) of 18 and 36 GHz frequency increases with an increase in SD up to a specific thickness, then saturates depending on snowpack conditions (Rango et al., 1979; Chang et al., 1987; Tedesco and
Narvekar, 2010). Hence, many studies of PMW SD inversion relied on empirical models derived using BTD between 18 and 36 GHz frequency TB observations (Chang et al., 1987; Saraf et al., 1999; Foster et al., 1997; Kelly et al., 2005, 2003; Das and Sarwade, 2008). Many of the empirical models for SD are developed by generalizing the snowpack parameters such as snow density, grain size, etc. (Chang et al., 1987, 1997; Kelly et al., 2003). However, these parameters dynamically vary with space and time. As a result, the applicability of many empirical SD models (Chang et al., 1987; Foster et al., 1997;
Aschbacher, 1989) outside their study region is not good, as evident from several studies (Dai et al., 2018; Wang et al., 2019, 2020; Saraf et al., 1999; Xiao et al., 2018). Further, many PMW studies have shown that the error in estimated SD using TB data varies with snow conditions (i.e., wetness, grain size, density), land cover, topography, ground SD, etc. (Dai et al., 2018; Tedesco and Narvekar, 2010; Tedesco et al., 2010; Kelly et al., 2002; Yang et al., 2021; Wang et al., 2010; Ansari et al., 2019). Different combinations of multifrequency PMW TB observations, snow information (i.e., snow cover fraction,
grain size, density), and auxiliary data such as topographical and landcover information are used in the PMW-based SD model development to account for these limitations (Dai et al., 2018; Wang et al., 2020, 2019). Many SD modeling approaches comprising static empirical linear (Chang et al., 1987; Saraf et al., 1999; Singh et al., 2012) and non-linear models (Wang et al., 2020, 2019), dynamic models (Tedesco et al., 2010; Grippa et al., 2004; Wei et al., 2021), snow emission models (Dai et al., 2018; Yang et al., 2021), machine learning algorithms (Xiao et al., 2018; Yang et al., 2020),
assimilation schemes (Kwon et al., 2017; Graf et al., 2006), etc., are developed using PMW TB, and auxiliary datasets for different regions.

Despite the significant progress in PMW based SD estimation, very few studies have been carried in the Indian WH using PMW data (Singh et al., 2012; Das and Sarwade, 2008; Saraf et al., 1999; Singh et al., 2015). WH being the tropical region, experiences significant changes in temperature leading to frequent melt-freeze snow events causing snow grain growth,
which introduces errors in the estimation of PMW SD (Singh et al., 2015). Further, the limited availability of in-situ SD observations, very high SD (i.e., > 1 m), and high spatiotemporal variability in snowpack characteristics pose numerous





constraints for PMW SD estimation in WH. Consequently, no studies were reported for PMW SD estimation in WH till 1999. For the first time, Saraf et al. (1999) estimated the average monthly SD using Scanning Multichannel Microwave Radiometer (SMMR) data onboard Nimbus-7 during 1979-1987 for the Sutlej valley region of Himalaya using the modified

Chang's model (Chang et al., 1992). However, the applicability of this model (Saraf et al., 1999; Chang et al., 1992) over the entire Himalaya cannot be justified as the model is developed using less amount of in-situ data (from 11 stations) where the stations are not distributed, and is not tested outside Sutlej basin. Singh and Mishra (2006) have proposed three empirical models using Advanced Scanning Microwave Radiometer for Earth (AMSR-E) data (horizontally polarized TB of 18.7 and 36.5 GHz) for SD estimation in the Pir-Panjal, Greater Himalaya, and Karakoram ranges of WH, respectively. Following this

study, Singh et al. (2007) used different empirical models for SD estimation using multifrequency Special Sensor Microwave/Imager (SSM/I) data (i.e., TB of 19, 22, 37, and 89 GHz during 1997-2002) over the Patseo region. However, these studies (Singh and Mishra, 2006; Singh et al., 2007; Saraf et al., 1999) have not provided any quantitative details about the accuracy of SD estimates and are not evaluated using independent SD observations.  Das and Sarwade (2008) used 18.7 GHz and 36.5 GHz horizontally polarized data from AMSR-E and modified the coefficients of Chang et al. (1987)'s model

to suit the Indian Himalaya. Their model (Das and Sarwade, 2008) has shown a mean absolute error (MAE) of 20.34 cm in SD estimates but failed to estimate SD above 60 cm.  Singh et al. (2012) have developed multiple empirical SD models for three SD classes, i.e., 1 to 5 cm, 5 to 50 cm, and 50 to 200 cm in Pir-Panjal, Greater Himalaya, and Karakoram regions of WH using TB data of different frequencies from SSM/I. Their approach (Singh et al., 2012) has used the scattering index to estimate snow cover, and TB thresholds for identifying the SD class and estimation of SD. In another study, Singh et al.

(2015) developed PMW SD models for the Dhundi and Patseo regions of Himalaya using data from ground-borne radiometers and in-situ observations. However, their models are developed using observations collected from only two field surveys, evaluated using a single day observation of AMSR-E TB data, and not tested spatiotemporally. Recently, Singh et al. (2020) developed an empirical algorithm for the Patseo region of the  MHZ using Advanced Microwave Scanning Radiometer 2 (AMSR2) 18.7 GHz and 36.5 GHz TB (i.e., during 2012-2016) and in-situ observations. They observed that

the estimated SD is very close to ground data with Root Mean Square Error (RMSE) of ~16 cm and MAE of ~13.9 cm.

Despite the development of various PMW SD models for Himalaya in the last two decades (1999 -2020), there are many constraints in the spatiotemporal estimation of SD for the WH region. Many of the previous studies for SD estimation in WH are carried out specifically for subregions of WH, such as the Sutlej basin, Dhundi, Patseo, etc. The PMW TB observations are affected by heterogeneity in snowpack properties, land cover, topography, etc. (Trujillo et al., 2007; Wang et al., 2010;

Che et al., 2016; Derksen, 2008; Foster et al., 2005).  However, previous studies (Das and Sarwade, 2008; Saraf et al., 1999; Singh et al., 2012; Singh et al., 2020, 2007; Singh and Mishra, 2006) have not accounted for the aforementioned variables. Further, the accuracy of SD retrievals from these models is also not evaluated with respect to the varying terrain and snow



parameters. The accuracy of operational PMW SD products available in the WH region, i.e., AMSR2 SD, has not been evaluated.

Additionally, the AMSR2 SD product and previous PMW SD models have course resolution and have limitations for their potential utility in various applications such as avalanche susceptibility, hydrological modeling, etc., especially at the regional scale. Considering these research gaps, in the current study, different linear and non-linear empirical models are developed to improve and estimate SD at high resolution, i.e., 500 m for different WH zones, using a multifactor approach. In this approach, multifrequency PMW observations from AMSR2 (during 2012-2019), terrain parameters, landcover

parameters, and Moderate Resolution Imaging Spectroradiometer (MODIS) derived snow cover product are statically correlated with the ground SD observations for the development and evaluation of the SD models. The accuracy of PMW multifactor SD models' estimates is compared with previous models and the AMSR2 SD product. Further, in this study, the SD retrievals accuracy is also analyzed with respect to different auxiliary parameters. The present study has the following three objectives:

1.  Development of multifactor SD models to estimate SD at high resolution for different WH zones.
    2.  Comparison and evaluation of the multifactor model, previous SD models for WH, and AMSR2 SD products in different WH zones.
    3.  Analysis of multifactor SD retrievals accuracy with respect to selected auxiliary variables.

Following this introduction section (section 1), the current article is organized as follows. The topographical and

geographical description of the study area is described in section 2. The details of the in-situ observations network and various remote sensing datasets used for model development and evaluation are also given in the same section. Following that, the methodology used in developing the multifactor model is presented in section 3. Subsequently, section 4 describes the performance of different multifactor models developed for the three WH zones, a comparison of the different SD models, and results from the analysis of multifactor SD model retrievals with respect to auxiliary parameters. Discussion and

summary are given in sections 5 and 6, respectively.

## 2. Study area and Datasets

The topographic and environmental conditions prevailing in WH are detailed in section 2.1. This study makes use of in-situ data from the snow monitoring network and various spaceborne data for the development of SD models for different WH zones. These datasets along with their sources, are listed in Table 1 and briefly discussed in the following subsections from

2.2 to 2.5.



## 2.1 Study area

Himalaya is the largest snow-covered territory outside the polar regions in the world (Gurung et al., 2011). The present study encompasses the entire WH, which is a significant portion of the Indian Himalaya, situated in the states of Jammu & Kashmir, Ladakh, and Himachal Pradesh (see Figure. 1). WH extends between longitudes from 73° 15' E to 79° 45' E,
latitudes from 30° 00' N to 39° N and covers an area of 3,60,866 km². WH is unique with its perennial snow-covered mountain peaks and seasonal snow-covered valleys. Approximately 65% of the terrain in WH is situated at an altitude of more than 3000 m above mean sea level (m.s.l.) and is underlain by extremely steep and rugged mountains. The high-altitude terrain and mountain topography influence both winter precipitation (caused by western disturbances) and monsoon precipitation patterns (Dimri and Dash, 2012). Due to prevailing topographical and weather conditions in WH, forest cover is
present only up to 3000 m (m.s.l.), and between 3000-4000 m (m.s.l.), thin vegetation consisting of shrubs and grass is present. Whereas above 4000 m (m.s.l.) altitude, vegetation is not present, and the landcover there is predominantly comprised of barren land with snow and ice. WH region generally receives snow from October to March; from April onwards, snow melt generates runoff contributing water to many rivers and streams within the region (Dimri and Dash, 2012; Sharma et al., 2014).

In this study, three WH zones defined based on the historical local meteorological and avalanche occurrence data (Sharma and Ganju, 2000) are used for developing multifactor SD models. The three WH zones are as follows: 1. Lower Himalayan Zone (LHZ), 2. Middle Himalayan Zone (MHZ), and 3. Upper Himalayan Zone (MHZ). The geomorphic and climate characteristics of different WH ranges are given in Table 2. The three zones differ in regional topographical and climatic conditions with varying elevations, temperatures, rainfall, snowfall, etc. The LHZ has a sub-tropical climate, and MHZ has a
temperate climate, while the UHZ has polar climatic conditions with the presence of permanent snow. Further, these zones have different timing and intensity of precipitation. LHZ has comparatively warmer conditions, with mean monthly temperatures varying between (-3° C to 18° C) than MHZ (-10° C to 14° C) and UHZ (-25° C to 0° C). As the latitude increase, the amount of precipitation deceases in WH. Negi et al. (2018) reported average winter precipitation (in terms of snow water equivalent) of ~ 804 mm, 549 mm, and 431 mm in the LHZ, MHZ, and UHZ, respectively, during 1991-2015. Further, the
snowpack persistence varies based on the local weather conditions, which mainly alter with elevation across the three WH zones.

## 2.2 Ground observatory stations data

In WH, Defence Geoinformatics Research Establishment (DGRE) (formally known as Snow and Avalanche Research Establishment) operates and maintains a network of 43 observatory stations (see Figure. 1) which measures daily in-situ SD
twice (i.e., forenoon and afternoon) along with other meteorological parameters such as temperature, rainfall, etc. Out of





total 43 stations, 16 stations are located in the LHZ, 13 in the MHZ, and 14 in the UHZ of WH. In the LHZ, MHZ, and UHZ observatories, elevation varies between 1652 to 3785 m (m.s.l.), 2440 to 4950 m (m.s.l.), and 3960 to 5995 m (m.s.l.), respectively. In this study, in-situ data comprising station name, date, latitude, longitude, and SD for the 43 stations is obtained for the snow period during 2012-13 to 2018-19. The in-situ data is grouped according to the WH zones for the

development of different multifactor SD models. The mean in-situ SD of stations varies between 11 to 256 cm in the LHZ, 23 to 136 cm in the MHZ, and 52 to 356 cm in the UHZ during the study period.

### 2.3 AMSR2 brightness temperature data and SD product

AMSR2 is a PMW sensor onboard the Japanese Aerospace Exploration Agency (JAXA)'s Global Change Observation Mission 1st - Water (GCOM-W1) SHIZUKU, launched in May 2012. It is a follow-on instrument to AMSR and AMSR-E

sensors, and records upwelling microwave emission from the earth's surface in 14 channels in the form of TB. AMSR2 TB observations are available in 7 frequencies (6.9, 7.3, 10.65, 18.7, 23.8, 36.5, and 89 GHz, hereafter referred to as 6, 7, 10, 18, 23 ,36 and 89 GHz) at two polarizations (horizontal and vertical) for ascending and descending orbit pass with a temporal resolution of 1 day. The multifrequency TB observations and SD product are re-gridded to 10 km spatial resolution (level-3 product) and are archived in the JAXA portal (http://gcom-w1.jaxa.jp). In many locations of WH, the temperatures

exceed $0^{\circ}$C from April to September, leading to snow melt (Negi et al., 2018; Sharma et al., 2014). The resulting wet snow can lead to saturation of PMW TB (Dong et al., 2005; Stiles and Ulaby, 1980; Tedesco et al., 2014), affecting the accuracy of SD estimates from PMW SD models. Therefore, in this study, the level-3 TB and SD product of ascending and descending orbital pass from the AMSR2 sensor are obtained for the snow/winter period (October to March) during 2012-2019 to develop and evaluate the SD models.

### 2.4 SRTM Digital elevation model

Topography affects the rate of snow accumulation, ablation, and redistribution. In the current study, Shuttle Radar Topography Mission (SRTM) digital elevation model (DEM) version 004 data at 90 m spatial resolution is used to account for the topographic effects in the SD model. SRTM DEM for the entire earth is generated using the interferometric synthetic aperture radar method (Farr et al., 2007; Jarvis et al., 2008) and can be downloaded from the web portal

(http:/srtm.csi.cgiar.org) in Geo-TIFF format. It has a minimum vertical accuracy of 16 m and RMSE of 9.73 m across the globe (Mukul et al., 2017). The SRTM DEM data is re-projected to the GCS-WGS-1984 coordinate reference system, then mosaiced, extracted, and resampled to 500 m spatial resolution. The elevation varies significantly across different WH ranges. LHZ and MHZ have less elevated topography than UHZ (See Figure. 1).



## 2.5 Daily MODIS cloud-free snow cover day products

SCD provides information regarding the persistence of snowpack and is useful in improving PMW SD estimates (Singh et al., 2016; Wang et al., 2019; Dai et al., 2018). In this study, daily could-free MODIS snow cover product (i.e., M*D10A1GL06) generated for high-mountain Asia (Muhammad and Thapa, 2020) at 500 m spatial resolution has been used to generate SCD product for the study area during the data period and can be downloaded from https://doi.org/10.1594/PANGAEA.918198. Sharma et al. (2014) and Singh et al. (2018) have generated and evaluated the

SCD maps for snow-covered Indian WH. SCD product depicts the number of consecutive days snow cover is present for a given pixel. In this study, SCD of the WH region is retrieved only during the study period i.e. from October to March for each year.

## 2.6 MODIS land cover product

The heterogeneity in landcover significantly impacts the amount of upwelling PMW radiation, and thereby affects the TB at

different frequencies for a given pixel. The effect of different types of land cover in PMW SD retrievals is investigated in many studies used (Friedl et al., 2002; Yu et al., 2012; Wang et al., 2016; Wang et al., 2019). In this study, MODIS Level 3 yearly land cover product (i.e., MCD12Q1) for the year 2019 is downloaded from https://www.ladsweb.modaps.eosdis.nasa.gov/ website at 500 m spatial resolution. MCD12Q1 product depicts land cover in 17 classes as per the International Geosphere-Biosphere Program (IGBP) system. These 17 classes are further regrouped into

4 categories i.e., bare land, grass land, forest, and water, which accounts for ~55.9%, 27.4%, 16.3%, and 0.29% of the total WH area in 2019, respectively. The reclassified landcover data has been used along with other datasets for development of multifactor SD models for different WH regions.

## 3. Methodology

Different steps followed for the development and validation of the multifactor SD model(s) are given in the following

subsections from 3.1 to 3.5. The general outline of the methodology adopted is shown in Figure. 2.

### 3.1 Data preprocessing

Different remote sensing datasets comprising PMW TB (from AMSR2), SRTM DEM, MODIS landcover product and MODIS SCD are used in the current study. These products are natively present in different spatial resolutions and coordinate systems. Hence, all remote sensing datasets are processed to match the spatial extent, coordinate system, and spatial

resolution. The AMSR2 TB data at different frequencies is resampled to 500 m spatial resolution using the nearest-neighbour



interpolation technique. The BTD is calculated between lower and higher frequency TB observations for each day during the study period.

Following the BTD calculation, the SRTM DEM product is re-projected, mosaiced, and resampled to 500 m spatial resolution. Different terrain parameters, such as slope, aspect, and surface roughness, are derived from the resampled DEM
product. SCD product is already available in 500 m spatial resolution. Therefore, it is processed only to match the extent and coordinate reference system (i.e., GCS-WGS 1984) of other datasets. Following the resolution and coordinate system matching process, for all DGRE observatory locations, the data from remote sensing products (i.e., TB, elevation, slope, ruggedness, geographical locations and SCD) is extracted for the winter period during 2012-13 to 2018-19. It is known that the forest cover intercepts the upwelling radiation from the ground underneath snowpack and causes uncertainty in the snow
depth estimates of PMW SD models (Che et al., 2008). Therefore, the forest cover fraction has been calculated using MODIS land cover type product (i.e., MCD12Q1) for a 10 km point buffer around each observatory site. The retrieved values are used to minimize the forest cover impact by dividing the brightness temperature observations with non-forest fraction (i.e., 1- forest fraction) as suggested by Foster et al., (1997).

In this study, the forenoon SD observations, descending pass AMSR2 TB data, terrain parameters (i.e., slope, aspect,
ruggedness), geographical locations and SCD are paired based on date and station location. These data are then checked for discrepancies such as missing values, incorrect values, outliers, etc. The samples containing such discrepancies are removed. After data preprocessing, a total of ~13,242 samples, with each sample comprising geographical location, TB, terrain parameters, and SCD, are retained. Using these samples, the data for five years snow period, i.e., from 2012-13 to 2016-17, is used to develop multifactor SD algorithms for different zones of WH. While the data of two years of snow period, i.e.,
from 2017-18 to 2018-19, is used to compare and validate the multifactor SD model results.

### 3.2 Identification of snow pixel

Along with snow cover, frozen ground, rainfall, and cold desert conditions affect the upwelling microwave emission from the earth's surface and impact PMW TB recorded by spaceborne sensors (Ferraro et al., 1996; Grody and Basist, 1996). Further, wet snow pixels and surface water bodies cause PMW absorption and reduce volume scattering from snow grains
(Stiles and Ulaby, 1980). Consequently, the inclusion of TB values from these pixels in the development and evaluation of the model results in large uncertainty in SD estimates (Tedesco et al., 2014; Dietz et al., 2012; Foster et al., 2005; Dong et al., 2005). Therefore, before developing SD algorithms, dry snow pixels must be segregated from other pixels. Grody and Basist (1996) have developed a decision tree to identify dry snow pixels from other scattering pixels using TB of different frequencies. Grody's decision tree makes use of different filters (see Figure. 3) based on the values of TB observations to



separate snow from non-snow pixels. This study uses multifrequency AMSR2 TB data with Grody's decision tree to identify snow pixels.

### 3.3 Selection of multifactor SD model parameters

Many of the initial PMW SD models have relied on TB from 18 and 36 GHz channels for estimating SD (Chang et al., 1987; Saraf et al., 1999; Das and Sarwade, 2008; Kelly et al., 2003; Chang et al., 1997). However, these models have limitations in

estimating shallow and deep snowpacks. The sensitivity of PMW TB to SD decreases once the SD reaches a threshold depth (Wang et al., 2019; Dai et al., 2018; Das and Sarwade, 2008; Kelly et al., 2003). TB of higher frequencies (i.e., 36 GHz, 89 GHz) saturate before lower frequencies (i.e., 10 GHz, 18 GHz) as SD increases. The lower frequency (i.e., 10 GHz) has the potential to retrieve deep snow cover, while the higher frequency (i.e., 89 GHz) can provide shallow snow information(Kelly et al., 2003). Therefore, the inclusion of higher frequencies (i.e., 89 GHz) and lower frequencies (i.e., 10, 23 GHz) are

investigated in many studies, which have resulted in improved SD estimates (Kelly et al., 2003; Wang et al., 2019; Xiao et al., 2020b; Wei et al., 2021). Hence, PMW TB of 10, 18, 23, 36, and 89 GHz are used in this study. Apart from single channel SD, 40 combinations of TB, i.e., BTD of different frequencies and polarizations, are also considered. Terrain parameters (i.e., elevation, slope, aspect, surface roughness), location (latitude, longitude), landcover, and SCD also affect characteristics of snowpack and PMW TB (Saydi and Ding, 2020; Sharma et al., 2014; Wang et al., 2010; Ansari et al.,

2019). Thus, overall, 57 parameters (i.e., TB – 10, BTD – 40, terrain parameters – 4, location – 2, SCD – 1) are considered in the process of SD model development. However, of the 57 parameters, it is likely that some parameters are redundant and do not necessarily add any value to the model. For example, TB of 10H and 10V have a correlation of 0.9 and using both TB10H, and TB10V is not useful and can cause additional problems due to multicollinearity. Further, the use of a large number of independent variables leads to a curse of dimensionality, which poses challenges in model development by

decreasing the model's interpretability, increasing the computational time and resources, and often overfitting (Velliangiri et al., 2019; Obaid et al., 2019). These problems can be addressed by performing optimal parameter selection for model development (Chandrashekar and Sahin, 2014). Optimal parameter selection reduces the data dimensionality and eliminates irrelevant data from the original dataset.

In this study, data from the snow period, i.e., between 2012-13 to 2016-17 of the entire WH, is considered for optimal parameter selection. To select the necessary parameters for the SD model, all 57 parameters are used independently with in-situ SD to develop single-parameter linear regression models. While developing these models, evaluation is carried out using the leave-one-outcross-validation (LOOCV) method (Webb et al., 2011) for screening necessary parameters. The LOOCV method is widely used by various researchers (Gusain et al., 2016; Joshi et al., 2017; Wang et al., 2019) to conduct





the validation of models and assess the model's accuracy. In LOOCV, the observational dataset is used to create n - number
of regression models (n is no of samples). In each of the n- models, a different testing sample is selected, and other
observation samples are used to develop the regression model. The overall performance of the model is calculated by
combining all predictions (for the omitted samples) from the n-models. The accuracy of the models is calculated using the
correlation coefficient (R) and RMSE. The results of LOOCV from the models developed with all 57 parameters (See
section 4.2) are analyzed to select the most valuable features for SD model development.

### 3.4 Development of SD models

This study implements four different regression models (i.e., linear, logarithmic, power, and reciprocal) to develop SD
models. Different WH zones, i.e., LHZ, MHZ, and UHZ, have different topographic, environmental, and snowfall
conditions. Hence, in this study, SD models are developed separately for each WH zones. Data from 2012-13 to 2016-17 is
used for the development of the different SD models. Further, out of 57 parameters, 13 parameters are selected from the
results of the LOOCV evaluation. These 13 parameters have a good correlation with in-situ SD and are used in developing
the multifactor SD models using four types of regression. The general form of the four types of regression models is given in
equations (1)-(4).

$$y = \alpha_0 + \alpha_1 x_1 + \alpha_2 x_2 + \cdots + \alpha_{i+1} x_i + c_i \tag{1}$$

$$y = \alpha_0 + \alpha_1 In x_1 + \alpha_2 In x_2 + \cdots + \alpha_{i+1} In x_i + c_i \tag{2}$$

$$y = b x_1^{\alpha_1} x_2^{\alpha_2} \underline{\cdots\cdots\cdots} x_i^{\alpha_i} e^{c_i} \tag{3}$$

$$y = \alpha_0 + \alpha_1 \frac{1}{x_1} + \alpha_2 \frac{1}{x_2} + \cdots + \alpha_{i+1} \frac{1}{x_i} + c_i \tag{4}$$

where, y is the ground observed SD values; $x_1$, $x_2$, ......., and $x_i$ are the screened parameters; $\alpha_0$, $\alpha_1$, $\alpha_2$, …, and $\alpha_i$ are the
regression coefficients of the multiparameter models; and $c_i$ is the offset constant.

### 3.5 Validation of SD model(s)

The multifactor SD models for different WH zones are validated using temporally independent in-situ SD observations
during 2017-18 and 2018-19. The accuracy of SD models' estimates is evaluated using standard regression metrics, i.e., R,
and RMSE. Additionally, the efficacy of the proposed multifactor SD models is analyzed by comparing the accuracy of the
multifactor model with regional (Das and Sarwade, 2008; Singh et al., 2020) and heritage SD model (Chang et al., 1987) for
different ranges of WH. Chang et al. (1987), Das and Sarwade (2008), and Singh et al. (2020) SD models are given in
Equations (5), (6), and (7), respectively.



The comparison is carried out by estimating SD from all these stated models during the study period, i.e., during 2017-18 to 2018-19.


$$\text{SD}_{\text{Chang et al. (1987)}} = 1.59 * (\text{TB18H} - \text{TB36H}) \qquad (5)$$

$$\text{SD}_{\text{Das et al. (2008)}} = 3.16 * (\text{TB18H} - \text{TB36H}) + 24.25 \qquad (6)$$

$$\text{SD}_{\text{Singh et al. (2020)}} = -7.58 * (\text{TB18V} - \text{TB36V}) + 233.71 \qquad (7)$$

Where, TB denotes the brightness temperature values, 18, and 36 indicate the frequency of TB in GHz, and V, H are the vertical and horizontal polarization, respectively.

Apart from the aforementioned comparative analysis, a random sample image from the study area for a single day
(February 3, 2019) is taken. Then the estimated SD over the selected area using the multifactor SD model(s) is spatially compared with AMSR2 operational products (See section 4.5). This spatial comparison helps in understanding how the developed multifactor SD model(s) differs from the AMSR2 operational SD products in representing SD information over WH. The magnitude of in-situ SD, terrain parameters, and SCD can significantly affect the accuracy of the PMW SD model in the study region. Therefore, the accuracy of operational AMSR2 SD products and multifactor SD models with respect to
varying ground SD, topographic elevation, and SCD is determined in different WH zones (see Section 4.6.).

## 4. **Results and analysis**

The insights from the analysis of in-situ SD observations in WH zones are reported in section 4.1. Following that, the results from the LOOCV evaluation of multiple parameters are given in section 4.2. The outcomes from the accuracy assessment and comparison of different PMW SD estimates are described in sections 4.3 and 4.4, respectively. The spatial comparison
of the high-resolution SD map from the multifactor model and AMSR2 products is shown in section 4.5. In section 4.6, the analysis of multifactor SD model performance with respect to different parameters is detailed.

### 4.1 Spatial analysis of the in-situ SD observations in WH ranges

DGRE has installed a total of 43 observatories in WH (Figure. 1.), which are situated at different elevations ranging between 1664 to 5995 m (m.s.l.). The mean of in-situ SD at each of the 43 DGRE stations is estimated for the winter period (October
to March) during 2012-13 to 2018-19 (See Figure. 4). The results indicate during the data period, the mean SD values varied between ~11 cm (elevation: 1664 m) to ~256 cm (elevation: 3160 m) in LHZ, ~21 cm (elevation: 3250 m) to ~136 cm





(elevation: 4950m) in MHZ, and ~49 cm (elevation: 3250 m) to ~365 cm (elevation: 5995 m) in the UHZ. The analysis also demonstrates that out of 43 manual stations, 4 stations have a mean SD between 11 to 50 cm, 18 stations have a mean SD between 50 to 100 cm, 7 stations have a mean SD between 100 to 150 cm, 5 stations have mean SD between 150 to 200 cm,

and remaining 4 have mean SD > 200 cm during the data period. Further, it is observed that out of 9 stations that have a mean SD greater than 150 cm, 5 are present in the UHZ.

The overall analysis of in-situ SD measurements indicates the mean and standard deviation ($\mu \pm \sigma$) are observed as ~121.5 cm ± 122.5 cm in the LHZ, ~85.9 cm ± 83.5 cm in the MHZ, and 176.6 cm ± 208.9 cm in UHZ, respectively. A higher mean SD is observed in the UHZ compared to the other two ranges. 95% of overall SD values in the LHZ are below 350 cm, with

the remaining 5% having SD between 350 cm and 650 cm. However, in the MHZ and UHZ, 95% of total SD observations are below 200 cm and 500 cm, respectively, with the remaining 5% ranging between 200 cm to 500 cm and 500 cm to 2030 cm.

## 4.2 SD parameters screening and evaluation over WH

The screening of parameters for the development of the multifactor SD model is carried out using the LOOCV approach

(See section 3.3). The data for winter (October to March) i.e., between 2012-13 to 2016-17, is used in the parameter selection process. The expressions for linear regression models developed with each parameter and regression metrics, i.e., RMSE, and R results obtained from LOOCV analysis, are shown in Table 3. In terms of geographical location, latitude has a higher correlation (i.e., 0.24) and lower RMSE (97.96 cm) than longitude. Among the terrain parameters, SCD has the highest R (i.e., 0.45) and the lowest RMSE (i.e., 90.27 cm), and is followed by elevation (R = 0.30 and RMSE = 96.12),

slope (R =0.26 and RMSE = 97.59), and ruggedness (R =0.25 and RMSE = 97.85), making it highly significant for the development of a multifactor SD models. The SD models built with TB observations from descending orbital passes have relatively higher correlation and lesser RMSE compared to those from ascending pass TB data when analyzed with in-situ SD. Therefore, only descending pass TB observations are used in the study.

Apart from the single channel PMW TB, 40 different combinations of descending pass orbital BTD are tested using

linear regression (in the LOOCV approach). The parameters that have the highest correlation and pass the F-test at a significance level of 0.001 are shown in Table 4. It is observed that descending pass BTD models exhibit higher correlation and accuracy metrics compared to single-channel descending pass models. From the overall results (R, RMSE), descending pass BTD parameter-based SD models have higher R (0.24 to 0.39) and lesser RMSE (91.63 to 93.92 cm) compared to single channel TB-based SD models, which have R (0.07 to 0.35) and RMSE (93.44 to 100.54 cm). Therefore, instead of

single channel TB, different BTD data from descending pass (See Table 4) are selected in the development of multifactor SD



models. Along with the eight descending BTD parameters, three terrain parameters (elevation, slope, ruggedness), latitude, and SCD are used in developing multifactor PMW SD models for the three WH ranges.

**4.3 Evaluation and comparison of different multifactor SD models in WH zones**

Regression analysis is carried out between the parameters (selected using LOOCV) and in-situ SD observations during 2012-
13 to 2016-17 (October to March month) in each WH zone. Different multiparameter SD models (i.e., linear, logarithmic, power, and reciprocal) are developed from regression analysis for all three zones of WH. The details of the developed models are given in Table 5. The results from the regression analysis indicate multifactor SD models developed with a power regression approach have a better fit with the in-situ data and outperformed other regression models with R (RMSE in cm) values i.e., 0.62 (49.17), 0.78 (37.72), 0.76 (55.12) in LHZ, MHZ, and UHZ, respectively.

The developed multifactor models in each WH zones are evaluated (with R and RMSE metrics) using temporally independent data during 2017-18 to 2018-19. Comparison of the four types of multifactor regression models in the LHZ, MHZ, and UHZ is carried out with the help of the Taylor diagram (See Figure. 5). The R, RMSE (in cm) metrics of (i) power, (ii) linear, (iii) logarithmic, and (iv) reciprocal models in different WH zones are as follows. LHZ (See Figure. 5a.): (i) 0.65, 22.7, (ii) 0.64, 29, (iii) 0.38, 52, (iv) 0.09, 121.3; MHZ (See Figure. 5b.): (i) 0.76, 19.2, (ii) 0.68, 22.8, (iii) 0.14, 41,
(iv) 0.47, 26.7; UHZ (See Figure. 5c): (i) 0.89, 22.6, (ii) 0.75, 36.5, (iii) 0.73, 36.9, (iv) 0.61, 43.2. The results from the comparison indicate that in all WH zones, the multifactor SD model developed using power regression has exhibited higher accuracy, i.e., better correlation and lesser RMSE compared to the models built using linear, logarithmic, and reciprocal regression approaches. Therefore, in each WH zones, multifactor SD model from power regression is used for estimating the PMW SD at 500 m spatial resolution.

**4.4 Comparative analysis of multifactor and other SD models in different zones of WH**

In order to compare the performance of different SD models in WH, the SD values are estimated using different models with the help of PMW and other auxiliary data during the study period (i.e., 2017-18 to 2018-19). Different models used in the comparative analysis are the multifactor SD model(s) from this study, regional SD models of WH (i.e., Das et al. (2008), Singh et al. (2020)), and the heritage SD model given by Chang et al. (1987). The estimated SD from each model is
compared with in-situ SD observations with in the respective WH zones to understand the accuracy of SD retrievals. Singh et al. (2020) model is proposed only for the MHZ. Therefore, it is not used for SD estimation in LHZ, and UHZ while doing comparative analysis.

In the LHZ of WH, both Chang's model and Das's model have poor correlation with in-situ SD and have shown RMSE (R) of 39.51 (-0.16), and 49.66 (-0.14) (See Figure. 6). Whereas the proposed multifactor SD model has shown a good





correlation with RMSE (R) of 32.87 (0.75). In the MHZ, Chang et al. (1987), Das et al. (2008), and Singh et al. (2020) have
        exhibited poor correlation with in-situ SD with R-values of 0.22, 0.21, and -0.22, respectively. Whereas, the proposed
        multifactor SD model has shown a good correlation with in-situ SD with an R-value of 0.65 (see Figure. 6). The RMSE is
        observed to be 36.32, 49.82, and 119.79 cm for Chang et al. (1987) model, Das et al. (2008) model, and Singh et al. (2020)
        model respectively. The proposed multifactor SD model has shown good accuracy with a lesser RMSE of 27.21 cm
compared to other SD models. The SD model proposed by Singh et al. (2020) for the MHZ is developed using data from a
        single observatory location. Hence, the Singh et al. (2020) model cannot represent the spatial variability of SD and shows
        significant errors with higher bias.

        Similar to the results observed in LHZ and MHZ, the Chang et al. (1987) model and Das et al. (2008) model have shown a
        poor correlation in the UHZ with R-values of 0.18 and 0.19, respectively. Whereas the multifactor SD model has shown a
good correlation with an R-value of 0.67 (See Figure. 6). The RMSE values are observed to be 60.95cm, 51.74 cm, and
        42.81 cm for Chang et al. (1987) model, Das et al. (2008) model, and proposed multifactor SD model in the UHZ. Overall
        results from the comparative analysis indicate that in each WH zones, the multifactor SD model has higher accuracy with
        good correlation (i.e., R) and lesser errors when compared with other models. Further, the developed model has exhibited
        better accuracy metrics in the MHZ compared to other zones in WH. The mean SD observed during the study period (i.e.,
2017-18 to 2018-19) in Pir-Panjal and UHZ are higher than the mean SD of the MHZ. Further, the LHZ has a forest canopy
        which can affect the PMW TB observations. Whereas, in the MHZ, most of the region is devoid of forest vegetation except
        for some patchy grass vegetation.  Hence it is expected to observe an increased error for SD models in LHZ and UHZ as
        compared to the MHZ.

### 4.5 Spatial comparison of SD from multifactor model and operational AMSR2 SD products: A case study

The spatial comparison of SD maps from AMSR2 SD products and the multifactor SD model is performed to understand the
        improvement of the AMSR2 multifactor SD model over the operational AMSR2 SD products in the WH region. For this
        purpose, the SD maps of operational AMSR2 SD products and multifactor SD model for WH zones for 3rd February 2019
        are considered (See Figure. 7). The SD spatial map at 500 m resolution for the WH zones is generated using multifactor SD
        model (See Figure. 7d). The AMSR2 ascending SD product, i.e., AMSR2_A (See Figure. 7b) and descending SD product,
i.e., AMSR2_D (See Figure. 7c) of the same region for the given day at 10 km resolution are also prepared.

        According to MODIS-derived SCA at 500 m resolution (See Figure. 7a) and DGRE observatories in-situ SD information,
        snow cover with varying thickness on 3rd February 2019 in WH zones. However, in both the AMSR2 SD products, i.e.,
        AMSR2_A and AMSR2_D, the majority of pixels have zero SD value, resulting in the underestimation of SD information
        by AMSR2 products. The maximum SD values observed in different products are as follows: AMSR2 ascending SD, 58 cm;





AMSR2 descending SD, 78 cm; Multifactor SD model, 476 cm; The multifactor SD model shows high heterogeneity in SD across the selected region in WH zones as compared to AMSR2 SD products. Further, the multifactor SD model offers good detail with regard to snow cover and provides SD data in the region at a high resolution of 500 m.

**4.6 Comparison of performance of Multifactor SD product, with operational AMSR2 SD product**

Though higher mean SD regions (i.e., LHZ, UHZ) have a higher error than lesser mean SD regions (i.e., MHZ), it is

important to assess how the SD products accuracy varies with changes in in-situ SD. Hence, in section 4.6.1, the operational AMSR2 SD products and AMSR2 multifactor model performances are analyzed in different SD classes. Further, it is also important to understand how the model's accuracy is affected with respect to different auxiliary parameters, i.e., topographical and landcover parameters. Therefore, the model accuracy is evaluated with respect to different topographical and land cover parameters, and the results are present in section 4.6.2.

**4.6.1 Analysis of operational AMSR2 SD products and multifactor SD model in different SD classes**

The AMSR2 SD products and multifactor SD model estimates are grouped into five SD classes, i.e., 0-25 cm, 25-50 cm, 50-75 cm, 75-100 cm, and >100 cm based on in-situ SD observations during 2017 -18 to 2018 -19. Along with the multifactor SD, the operational AMSR2 SD product (i.e., from both ascending and descending pass data) is also analyzed in the SD classes by comparing with in-situ SD observations. RMSE of each SD class is calculated to evaluate the accuracy of SD

estimates. Other models (i.e., Chang et al., (1987), Das et al. (2008), and Singh et al. (2020)) were not considered in this analysis as they were not operational SD models. The effect of variation in ground SD on the accuracy of the AMSR2 multifactor SD model and AMSR2 SD products is shown in Table 6.

The results indicate that the magnitude of RMSE error of AMSR2 SD products and multifactor SD model increased with an increase in SD. When in-situ SD < 25 cm, AMSR2 SD products have shown relatively lesser error in all three zones

compared to the developed multifactor SD model. However, the observed error in this class (i.e., 0-25 cm) is still large and varies between 11 – 15 cm in the AMSR2 SD product and 14 – 27 cm in the multifactor SD model across the three zones. Whereas for classes with in-situ SD > 25 cm, the proposed multifactor SD model has low RMSE than both AMSR2 products in all the zones of WH. This analysis clearly shows that for shallow snow regions in WH, operational AMSR2 products can be used. However, for deep and moderate snow regions, the AMSR2 SD products show a large error. However, in the WH

regions, out of 43 stations, only four stations have a mean SD < 25 cm, and for the remaining stations mean in-situ SD is more than 25 cm during the study period. Hence AMSR2 SD products are less useful for spatial monitoring of SD in WH. Though the developed AMSR2 multifactor model has shown higher error when in-situ SD < 25 cm, it is more useful for the WH region as the RMSE error is lesser when SD > 25cm.





Overall, the multifactor SD model in MHZ has a low RMSE compared to the LHZ and UHZ in all SD classes. This could be
due to prevailing dry snow conditions, lesser mean SD, and the absence of forest in this range, which can act in favour of the
PMW SD algorithms to retrieve better SD estimates from PMW TB. Whereas higher temperatures, moist snow conditions,
forest vegetation in LHZ, and deep snow conditions in both LHZ and UHZ can deter the accuracy of the AMSR2 multifactor
SD model in these regions by affecting the TB observations.

### 4.6.2 Multifactor model performance analysis with respect to auxiliary parameters

Among all the factors considered in the AMSR2 multifactor SD model development, elevation and SCD have good
heterogeneity across the stations in each WH zone. The other terrain factors, such as slope, landcover have not much
variation and are similar for many of the stations with in a WH zone (See Figure. 8.). Though the large variation in landcover
is observed across the entire WH region, in the LHZ majority of stations are surrounded by forest cover. Whereas in the
MHZ, stations are mainly over grassland and barren land. The UHZ is devoid of vegetation, and all stations are present over
barren land and glacier. Thus, the variation of landcover with in a range is not significant. Therefore, in this section, only the
effect of varying elevation and SCD on the accuracy of SD from the AMSR2 multifactor SD model and operational AMS2
SD products is evaluated.

Slope and SCD are divided into different classes considering the overall variation in the WH region. Within these classes,
the SD retrievals from the AMSR2 products and multifactor model are compared with in-situ SD measurements during the
winter period of between 2017-18 to 2018-19. The accuracy of model estimates in each class is evaluated by calculating
RMSE (See Table 7). The RMSE error associated with each station for different factors (i.e., elevation, slope, SCD, and
landcover) is depicted in Figure. 8. The results indicate, in LHZ and UHZ, with an increase in elevation, RMSE increased for
both AMSR2 products, and multifactor model. Whereas in the MHZ, there is no specific trend in the variation of accuracy
with respect to elevation. The RMSE error (in cm) variation across all elevation classes for the multifactor SD model,
AMSR2 ascending SD, and AMSR2 descending products in different WH regions is as follows: LHZ, 21.38 – 47.27 cm,
23.05 – 113.21 cm, and 18.44 – 93.72 cm; MHZ, 17.82 – 54.79 cm, 39.10 – 107.72 cm, and 37.98 – 103.38 cm; UHZ, 11.73
– 126.13 cm, 17.71 to 188.67 cm, and 19.50 – 182.12 cm. Though overall RMSE variation is high across the different
elevation classes, both AMSR2 SD products have similar RMSE errors for any given elevation class within each WH zone.
However, the multifactor SD model has lower RMSE compared to both AMSR2 SD products for elevation classes across the
three WH zones. Other than elevation, the amount of snowfall and snow conditions vary widely with SCD across the
different WH zones. This can lead to varying accuracy trends in SD retrievals for a given factor in different WH zones.

The SD, in general, increases with an increase in SCD, affecting the PMW SD retrieval from different PMW SD models. In
WH, across all regions, the RMSE values of the AMSR2 ascending product, AMSR2 descending product, and the

multifactor SD model increased with an increase in SCD. However, the RMSE of the multifactor SD model is significantly
low compared to the AMSR2 SD products in all SCD classes in each WH zone. The SCD variation at the end of the snow
year (September 30, 2013), along with RMSE errors of different stations calculated for the time period i.e., 2017-18 to 2018-
19, is represented in Figure. 10(d). The RMSE variation associated with SCD classes for multifactor SD model, AMSR2
ascending product, and AMSR2 descending product in different WH ranges are as follows: LHZ, 25.18 – 61.80 cm, 47.32 –
158.11 cm, and 45.47 – 140.59 cm; MHZ, 19.71 – 70.92 cm, 33.53 – 140.03 cm, and 31.61 – 137.66 cm; UHZ, 83.40 –
122.22 cm, 97.63 – 205.89 cm, and 92.54 – 204.15 cm.

## 5. Discussion

The Indian WH has the highest elevation mountain peaks in Asia that segregate the plane regions of the Indian subcontinent
from the Tibetan Plateau with a mean elevation of ~3116 m (m.s.l.). WH consists of three mountain zones with diverse land
cover and demonstrates high spatial viability in SD due to complex terrain topography and climate conditions. Precise SD
information in the WH regions is very challenging to retrieve using spaceborne PMW remote sensing due to coarse spatial
resolution. Therefore, in this article, multiparameter (i.e., geographical location, terrain parameters, SCD, and BTD) based
PMW SD models that estimate SD at 500 m spatial resolution are developed for each WH zone. The discussion about the
performance of SD models and factors affecting the multifactor SD model is given in the following sections, 5.1 and 5.2,
respectively.

### 5.1 SD models' performance

Four multifactor SD models are developed for each WH zone using different regression approaches (i.e., linear, logarithmic,
reciprocal, and power). These models are compared with regional SD models, Chang's SD model, and operational AMSR2
SD products. The overall analysis of the results indicates that the power regression-based multifactor SD model has higher
accuracy compared to other multifactor SD models, regional approaches, Chang's model, and AMSR2 SD products in all
WH regions. However, AMSR2 SD products have shown comparable to better accuracy (i.e., similar to the multifactor SD
model) under shallow snow conditions (SD < 25 cm). Nevertheless, once SD exceeds 25 cm, the performance of AMSR2 SD
products declined considerably (See Table 7). Further, AMSR2 SD products have a large amount of missing data over the
WH region, rendering its poor utility for various regional applications.

In general, with an increase in SD, the accuracy of multifactor models declined in all WH zones. However, the accuracy of
developed multifactor SD models is distinct for a given SD class in different WH zones. This is because the spatial
distribution of snowfall and snow characteristics (i.e., SD, snow wetness, density, etc.) are not uniform at different





geographic locations of observatories distributed across the three WH zones. The SD model developed in the MHZ has shown better accuracy metrics than those developed for LHZ and UHZ. Different factors affecting the performance of the multifactor SD model are discussed in the following section, 5.2.

**5.2 Factors affecting the performance of multifactor SD model**

A total of 72 parameters comprising multifrequency PMW TB, BTD, terrain parameters, and SCD are screened using LOOCV to determine the suitable factors to develop PMW multifactor SD model. Finally, the SD algorithm is developed only using the selected parameters, i.e., geographical location parameters (latitude), terrain parameters (elevation, slope, and ruggedness), SCD and BTDs (36H89V, 36V89V, 10V23H, 23H89V, 10V18V, 10H23H, 10H18H, 18H89V). In the present study, different combinations of frequencies in vertical and horizontal polarization have been used to estimate shallow to deep snow across different zones of WH. Only descending pass PMW observations are employed in this study to avoid the problems pertaining to wet snow, which is more prominent during the ascending pass. Among the different factors (i.e., other than PMW data) evaluated using LOOCV, SCD has shown a relatively strong correlation (i.e., R = 0.45) with in-situ SD observations. Higher SCD generally indicates longer snow persistence which leads to an increase in snow accumulation, whereas lesser SCD indicates absence/melt of snow which leads to lesser SD.

Apart from SCD, terrain parameters, i.e., elevation, slope have an impact on the spatial distribution of SD within an area (Saydi and Ding, 2020; Trujillo et al., 2007; Sharma et al., 2014) and have shown a strong correlation with in-situ SD with R of 0.30, 0.26 respectively (See Table 3). Further, the topographic conditions can affect the reallocation of PMW radiation due to variations in the direction of polarization and local incidence angles (You et al., 2011), altering the TB values. The higher elevation regions (i.e., UHZ) of WH experience cold conditions, which aids snow in accumulating. Therefore, the snowfall is preserved for most of the winter in higher mountain areas of MHZ and UHZ, leading to higher SD in these regions. The accuracy of PMW SD models varies with the magnitude of in-situ SD, as evident from the current study, as well as from many previous studies (Xiao et al., 2020b, a; Dai et al., 2018). However, there are many other factors (such as land cover, snow wetness, grain size, etc.) that can affect the accuracy of SD retrievals (Dong et al., 2005; Foster et al., 2005; Tedesco et al., 2010; Kurvonen and Hallikainen, 1997; Ansari et al., 2019). It is important to note that the land cover and snow conditions alter considerably from one range to another in WH. Therefore, for any given parameter (such as SCD, elevation, slope, etc.), the accuracy trend of multifactor SD model estimates is not uniform when compared between different zones. The LHZ has forest vegetation, higher temperature, and higher mean SD (compared to Greater Himalaya). These conditions decline the accuracy of the SD estimates from the PMW multifactor SD model than the estimates observed in MHZ. Whereas, in MHZ, the absence of forest cover, relatively less mean SD (compared to both LHZ, and UHZ), stable snow conditions pose relatively better conditions for SD estimation using PMW data. Therefore, compared to other ranges,



in MHZ, the multifactor SD model has shown improved accuracy. Whereas, in the UHZ, higher SD is present due to which PMW signal saturates; hence larger errors in SD are observed for the multifactor SD model in this region. Thus location (i.e., latitude), land cover, elevation, SCD, and magnitude of in-situ SD altogether play an important role in the accuracy of
multifactor model SD estimates in the WH region.

## 7. Summary and Conclusions

The contrasting climate and snow conditions prevailing in Western Himalaya (WH) zones present new challenges in accurate snow depth (SD) retrievals using spaceborne Passive Microwave (PMW) remote sensing. The limited access to in-situ SD data, rugged topography, and inclement weather resulted in fewer SD studies over the WH region. The analysis of
operational PMW SD products (i.e., AMSR2 SD) available at the hemisphere scale also indicates a significant amount of missing data in WH.

Considering the topographical and climate conditions, the WH region is divided into three zones, i.e., LHZ, MHZ, and UHZ, in this study. In the mountainous region, the topography parameters, i.e., elevation and slope, affect the snow precipitation and its persistence. Therefore, the current study presents a multifactor approach that considers a number of auxiliary
parameters, such as terrain (elevation, slope), location, SCD, etc., for developing the PMW SD model using multifrequency AMSR2 TB observations. Different regression approaches (i.e., linear, logarithmic, reciprocal, and power) are used for developing the multifactor SD model to estimate SD at 500 m spatial resolution in each WH zone. The overall results indicate power regression performed better compared to other tested approaches in all zones. The results of the multifactor model from power regression are evaluated by comparing the SD estimates with ground SD, other SD products, and PMW
models. The results indicate under deep snow (>25 cm) conditions the developed multifactor model has shown higher accuracy compared to the AMSR2 operational SD product and other SD models. However, the accuracy of SD from the multifactor model is affected by variations in auxiliary parameters such as SCD, elevation, etc. With an increase in SCD, the SD increased in each WH zone. Further, the RMSE error associated with SD is also increased alongside SCD and SD in each WH zone. The MHZ has stable snow conditions with relatively less thick snowpack. Therefore, the multifactor SD model in
this region has shown improved accuracy for a given SD class compared to other WH zones. Overall, the proposed multifactor SD models for WH zones have demonstrated substantial improvement in estimating SD compared to the operational AMSR2 SD product, heritage SD model, i.e., Chang's model, and previous models developed within WH zones. Though multifactor SD model has outperformed other models and products, there is still scope for improving PMW SD estimates in WH. The developed model(s) has shown poor performance compared to AMSR2 products when SD <25 cm.
This can be possibly attributed to wet snow conditions prevailing in the early winter, i.e., when SD will be shallow. Further,

the inclusion of snowpack characteristics such as snow grain size, wetness, density data during the model development can improve the accuracy of SD estimates. The available in-situ SD observations are very limited considering the high spatiotemporal variability of SD in this region. Therefore, there is an immediate need of expanding the in-situ network of monitoring stations, and field-based studies to determine the first-hand knowledge of snowpack information in WH region.

Additionally, the potential of machine learning models for SD estimation in WH is not explored and can be investigated for the betterment of SD.

**Author contributions**

Dhiraj Kumar Singh, Tanniru Srinivasarao, and RAAJ Ramsankaran designed the working methodology of the study. Kamal Kant Singh and Harendra Singh Negi assisted in curating the in-situ data and analysis of the results and discussions. Dhiraj

Kumar Singh and Tanniru Srinivasarao have executed the overall work and wrote the manuscript. All the authors were involved in manuscript revisions. RAAJ Ramsankaran supervised the work.

**Data availability**

The in-situ dataset used in the study can be collected upon request and subject to approval from Defence Geoinformatics Research Establishment, Chandigarh, India.

**Competing interests**

The authors declare that they have no conflict of interest.

**Acknowledgments**

This work was sponsored by Defence Geoinformatics Research Establishment (DGRE), DRDO, under CARS project. The authors are grateful to the Director DGRE and Chairman CARS review committee of DGRE for their guidance and technical
suggestions during the different milestones of the project. The support of DGRE team is also appreciated during field visits. We are also thankful to JAXA for providing AMSR2 data for carrying out this work.



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



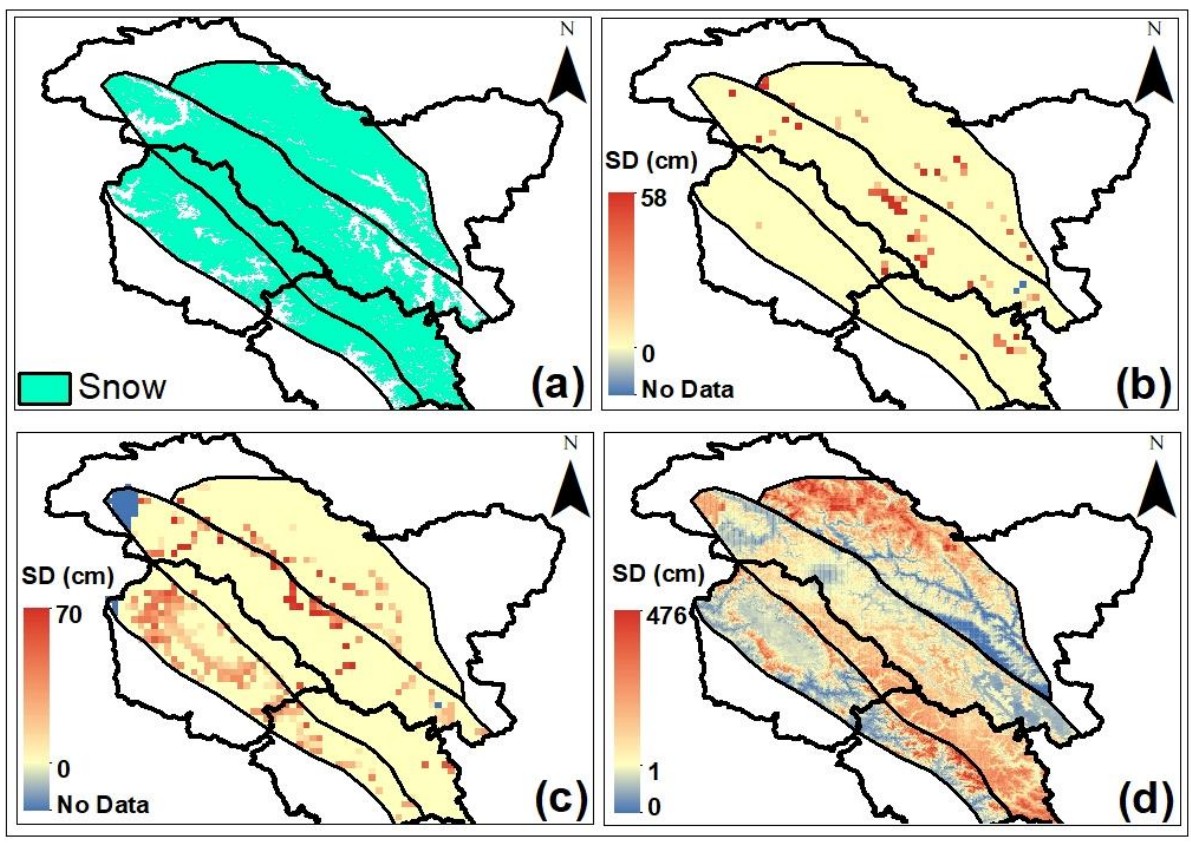

**Figure. 7: Spatial map of SD variation on 3rd Feb 2019. (a) MODIS SCA, (b) AMSR2_A SD product map at 10 km, (c) AMSR2_D SD product map at 10 km, and (d) multifactor models SD map at 500 m.**



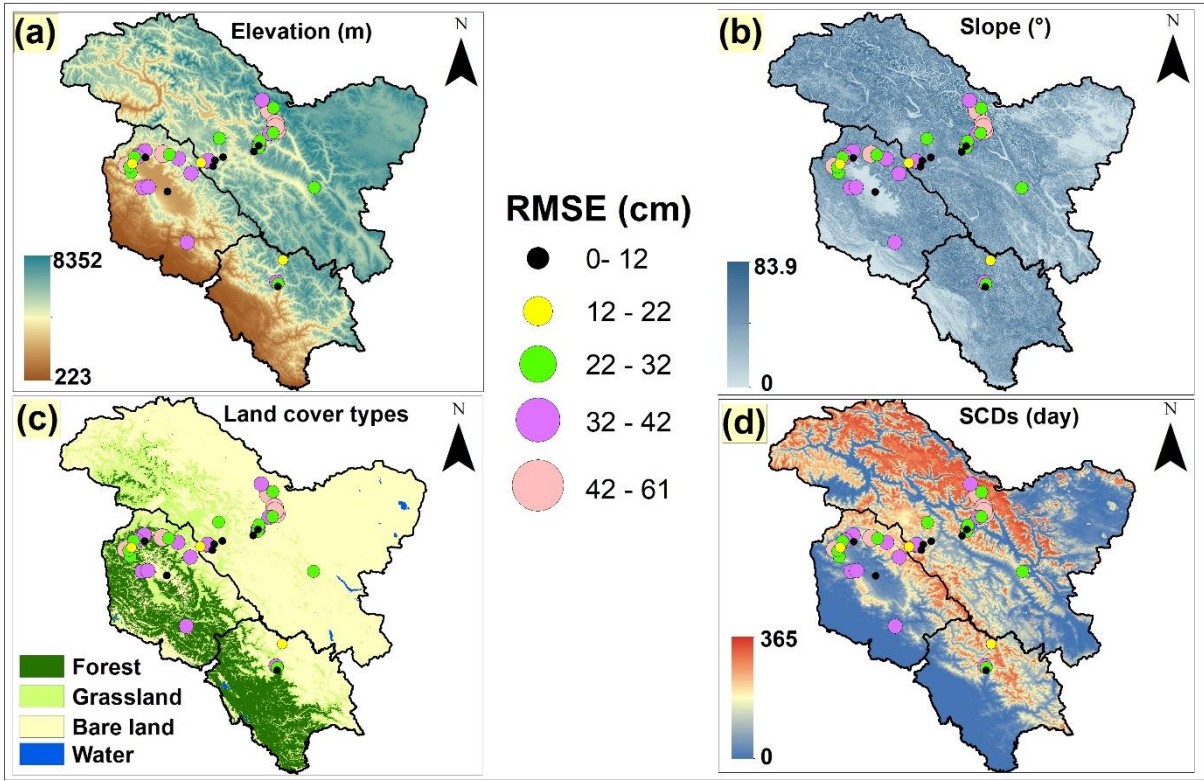

**Figure. 8. Spatial distribution of RMSE of multifactor SD model for varying (a) elevation, (b) slope, (c) land cover types, and (d) SCDs along the 43 ground stations**





**Table 1. Sources of in-situ, remote sensing datasets and their application in the present study**

| Data | Source | Role/Applications |
|------|--------|-------------------|
| In-situ snow depth data | DGRE Data Centre, Chandigarh, India | Development and validation of multifactor SD models |
| AMSR-2 Brightness temperature Snow depth product | http://gcom-w1.jaxa.jp/ | Development, validation, and comparison of multifactor SD models |
| MODIS Land cover data (MCD12Q1) | www.ladsweb.nascom.nasa.gov | Development of SD models |
| Daily MODIS cloud-free snow cover product | https://doi.org/10.1594/PANGAEA.918198 | Development of models and model performance analysis |
| Digital Elevation Model | http://srtm.csi.cgiar.org | Development and model performance analysis |






**Table 2. Geomorphic characteristics of the WH zones**

| Characteristics/Ranges | Lower Himalaya | Middle Himalaya | Upper Himalaya |
|---|---|---|---|
| **Area (km²)** | 41,107 | 73,951 | 38,441 |
| **Elevation** | 1500-4800 | 1500 -5700 | 1800-8100 |
| **Climate Type** | Sub-tropical | Temperate | Polar |
| **Winter snow fall (from -to)** | High (Dec-Mar) | Moderate (Oct-Apr) | Scanty (Entire year) |
| **Frequency of high-intensity snowfall events** | Occasional | Frequent | Occasional |
| **Vegetation cover presence** | Forest: < 3000 m (m.s.l.) Grass: 3000 - 4000 m (m.s.l.) | Grass: <3000 m (m.s.l) | ---- |
| **Snowpack persistence up to** | Early spring | Spring | Summer |




**Table 3. Results of LOOCV evaluation for SD models developed using single parameters**

| Parameters | Independent Variable (x) | Linear Regression Model | RMSE (cm) | R |
|---|---|---|---|---|
| Geographical location | Latitude | y = 32x – 1046.81 | 97.96 | 0.24 |
| | Longitude | y = 21x – 1573.72 | 99.30 | 0.17 |
| Terrain | Elevation | y = 0.029x – 45.81 | 96.12 | 0.30 |
| | Aspect | y = - 0.16x + 109.58 | 99.75 | 0.14 |
| | Slope | y = - 3.42x + 119.67 | 97.59 | 0.26 |
| | Ruggedness | y = - 0.31x + 128.15 | 97.85 | 0.25 |
| Cloud-free snow product | SCD | y = 1.14x – 19.58 | 90.27 | 0.45 |
| Brightness temperature (Ascending data) | 10H | y = - 4.5x + 1210.78 | 106.28 | 0.34 |
| | 10V | y = -5.77x + 1553.53 | 105.58 | 0.36 |
| | 18H | y = - 3.9x + 1047.89 | 104.59 | 0.39 |
| | 18V | y = - 4.8x + 1293.91 | 103.74 | 0.40 |
| | 23H | y = - 3.59x + 963.28 | 105.22 | 0.37 |
| | 23V | y = - 4.13x + 1109.64 | 104.65 | 0.38 |
| | 36H | y = - 2.1x + 587.17 | 108.81 | 0.28 |
| | 36V | y = - 2.22x + 620.55 | 109.08 | 0.27 |
| | 89H | y = - 0.24 + 161.87 | 113.29 | 0.03 |
| | 89V | y = - 0.08x – 90.99 | 113.35 | 0.01 |
| Brightness temperature (Descending data) | 10H | y = - 3.2x + 858.65 | 94.28 | 0.35 |
| | 10V | y = - 4.01x + 1071.98 | 93.96 | 0.36 |
| | 18H | y = - 2.76x + 741.75 | 93.96 | 0.36 |
| | 18V | y = - 3.4x + 910.08 | 93.44 | 0.38 |
| | 23H | y = - 2.67x + 706.15 | 94.05 | 0.36 |
| | 23V | y = - 3.05x + 811.49 | 93.74 | 0.37 |
| | 36H | y = - 1.78x + 480.19 | 96.38 | 0.29 |
| | 36V | y = - 1.9x + 522.28 | 96.42 | 0.29 |
| | 89H | y = - 0.58x + 207.88 | 100.29 | 0.10 |
| | 89V | y = - 0.45 + 181.84 | 100.54 | 0.07 |

y = SD (cm)





**Table 4. BTD SD model relation with SD and evaluation using LOOCV method**

| Parameters | Independent Variable (x) | Linear Regression Model | RMSE (in cm) | R |
|---|---|---|---|---|
| | BTD (36H-89V) | y = - 2.24x + 107.05 | 91.63 | 0.39 |
| | BTD (36V-89V) | y = - 2.16x + 81 | 92.24 | 0.37 |
| | BTD (10V-23H) | y = 4.12x + 31.05 | 92.45 | 0.35 |
| Brightness Temperature Difference (Descending data) | BTD (23H-89V) | y = - 1.78x + 122.17 | 92.46 | 0.36 |
| | BTD (10V-18V) | y = 7.43x + 52 | 92.58 | 0.25 |
| | BTD (10H-23H) | y = 4.12x + 56 | 93.78 | 0.20 |
| | BTD (10H-18H) | y = 5.66x + 58 | 93.47 | 0.21 |
| | BTD (18H-89V) | y = - 1.61x + 122.34 | 93.92 | 0.24 |

Note: y = SD (cm)






**Table 5. Multifactor SD model regression coefficient for WH zones during 2012-2017 (October to March)**

| WH zones | Model type | Models | R (RMSE) |
|---|---|---|---|
| **Lower Himalayan Zone** | Linear | $y = -51.16 - 0.09x_1 + 0.08x_2 + 0.94x_3 - 0.24x_4 + 1.15x_5 - 0.37x_6 + 5.21x_7 - 1.6x_8 - 0.40x_9 + 1.14x_{10} + 0.001x_{11} - 0.45x_{12} - 4.07x_{13}$ | 0.59 (64.14) |
| | Logarithmic | $y = -528.08 - 733.37\ln(x_1) + 678.83\ln(x_2) + 8.80\ln(x_3) - 230.65\ln(x_4) + 84.84\ln(x_5) + 19.32\ln(x_6) - 0.82\ln(x_7) + 30.21\ln(x_8) + 22.66\ln(x_9) - 11.51\ln(x_{10}) - 132.43\ln(x_{11}) + 12.23\ln(x_{12}) - 38.37\ln(x_{13})$ | 0.45 (81.12) |
| | Power | $y = 4.49 * 10^{-26}x_1^{10.12}x_2^{4.93}x_3^{1.28}x_4^{-2.29}x_5^{0.59}x_6^{-0.25}x_7^{0.56}x_8^{-2.79}x_9^{0.04}x_{10}^{0.22}x_{11}^{1.70}x_{12}^{0.001}x_{13}^{-0.68}$ | 0.62 (49.17) |
| | Reciprocal | $y = 487.86 - 5073.85/x_1 - 835884.29/x_2 - 619.77/x_3 + 45566.55/x_4 - 226.44/x_5 + 1.04/x_6 + 2/x_7 + 24.12/x_8 + 0.19/x_9 - 1.13/x_{10} + 69.02/x_{11} - 0.37x_{12} - 75.87/x_{13}$ | 0.49 (78.11) |
| **Middle Himalayan Zone** | Linear | $y = 1285.89 - 34.66x_1 + 0.001x_2 + 1.57x_3 - 0.05x_4 + 1.50x_5 - 4.10x_6 + 7.73x_7 - 1.77x_8 + 2.90x_9 - 1.39x_{10} + 0.001x_{11} + 4.70x_{12} - 8.84x_{13}$ | 0.69 (42.04) |
| | Logarithmic | $y = 2281.76 - 1364.46\ln(x_1) + 0.24\ln(x_2) + 27.81\ln(x_3) - 35.38\ln(x_4) + 120.18\ln(x_5) + 4.53\ln(x_6) + 23.03\ln(x_7) - 49.38\ln(x_8) + 15.60\ln(x_9) + 26.05\ln(x_{10}) - 83.25\ln(x_{11}) + 28.42\ln(x_{12}) - 124.63\ln(x_{13})$ | 0.62 (51.11) |
| | Power | $y = 3.7 * 10^{13}x_1^{-7.72}x_2^{0.03}x_3^{0.03}x_4^{0.07}x_5^{1.09}x_6^{-0.11}x_7^{0.57}x_8^{-0.01}x_9^{0.05}x_{10}^{-0.12}x_{11}^{-0.73}x_{12}^{0.19}x_{13}^{-1.51}$ | 0.78 (37.72) |
| | Reciprocal | $y = -26.58 + 4283.25/x_1 - 2060.51/x_2 + 20.08/x_3 - 6150.16/x_4 - 268.45/x_5 - 0.04/x_6 - 0.35/x_7 + 23.07/x_8 - 0.19/x_9 - 0.03/x_{10} + 2.63/x_{11} - 0.16/x_{12} + 5.72/x_{13}$ | 0.66 (45.72) |





| WH zones | Model type | Models | R (RMSE) |
|---|---|---|---|
| Upper Himalayan Zone | Linear | $y = -3754.98 + 99.59x_1 + 0.07x_2 - 1.25x_3 + 0.06x_4 + 1.23x_5 + 10.46x_6$ $+ 0.001x_7 + 4.90x_8 - 17.63x_9 + 19.77x_{10} - 6.58x_{11}$ $+ 6.06x_{12} - 16.11x_{13}$ | 0.74 (58.07) |
| | Logarithmic | $y = -8280.01 + 4430.22\ln(x_1) + 378.71\ln(x_2) - 102.97\ln(x_3)$ $+ 68.20\ln(x_4) + 116.51\ln(x_5) + 31.14\ln(x_6) - 6.29\ln(x_7)$ $- 60.95\ln(x_8) - 16.924\ln(x_9) + 42.65\ln(x_{10})$ $- 9.88\ln(x_{11}) + 15.42\ln(x_{12}) - 74.45\ln(x_{13})$ | 0.68 (69.08) |
| | Power | $y = 6.4 * 10^{-53} x_1^{26.26} x_2^{2.91} x_3^{-0.20} x_4^{0.62} x_5^{0.64} x_6^{0.11} x_7^{-0.18} x_8^{-0.31} x_9^{-0.13} x_{10}^{0.23} x_{11}^{0.10} x_{12}^{0.001} x_{13}^{0.44}$ | 0.76 (55.12) |
| | Reciprocal | $y = 5308.54 - 170987.89/x_1 - 1378163.61/x_2 + 121.52/x_3 - 5412.19/x_4$ $- 197.55/x_5 + 0.08/x_6 + 0.51/x_7 - 0.45/x_8 - 0.60/x_9$ $+ 1.15/x_{10} - 0.26/x_{11} - 0.09/x_{12} - 63.24/x_{13}$ | 0.33 (59.61) |






**Table 6. RMSE of operational AMSR2 SD products and the proposed multifactor SD model across different WH zones in different in-situ SD classes**

| WH zones | Model | Snow depth class (in cm) | | | | |
|---|---|---|---|---|---|---|
| | | 0-25 | 25-50 | 50-75 | 75-100 | >100 |
| **Lower Himalayan Zone** | AMSR2_A | 14.60 | 38.88 | 62.31 | 88.63 | 159.16 |
| | AMSR2_D | 14.35 | 35.68 | 58.65 | 85.75 | 152.84 |
| | Multifactor SD model | 27.64 | 21.62 | 37.27 | 40.48 | 63.73 |
| **Middle Himalayan Zone** | AMSR2_A | 13.67 | 35.97 | 61.39 | 86.34 | 200.25 |
| | AMSR2_D | 18.00 | 33.45 | 58.59 | 82.77 | 193.76 |
| | Multifactor SD model | 20.99 | 20.29 | 27.84 | 41.95 | 81.04 |
| **Upper Himalayan Zone** | AMSR2_A | 13.23 | 38.13 | 61.58 | 87.84 | 375.61 |
| | AMSR2_D | 14.34 | 36.93 | 60.19 | 86.69 | 372.67 |
| | Multifactor SD model | 37.12 | 41.54 | 38.62 | 40.12 | 161.01 |





**Table 7. RMSE (cm) variation of AMSR2 SD products, and multifactor SD model for different elevation, and SCD classes across three WH zones (for snow period during 2017-18 to 2018-19)**

| Parameter | Parameter range | Elevation (in m) | | | | | | | SCDs (in Days) | | | |
|---|---|---|---|---|---|---|---|---|---|---|---|---|
| | | 2000-2500 | 2500-3000 | 3000-3500 | 3500-4000 | 4000-4500 | 4500-5000 | >5000 | 0 - 30 | 30 - 60 | 60 - 90 | > 90 |
| **Lower Himalayan Zone** | Mean in-situ SD | 18.44 | 63.09 | 93.72 | - | - | - | - | 37.32 | 84.33 | 96.95 | 140.38 |
| | AMSR2_A | 23.05 | 76.97 | 113.21 | - | - | - | - | 47.32 | 101.59 | 114.23 | 158.11 |
| | AMSR2_D | 26.51 | 78.43 | 107.99 | - | - | - | - | 45.47 | 95.71 | 111.62 | 140.59 |
| | multifactor model | 21.38 | 41.13 | 47.27 | - | - | - | - | 25.18 | 42.12 | 54.22 | 61.8 |
| **Middle Himalayan Zone** | In situ SD | 80.53 | 59.96 | 31.8 | 69.42 | 83.02 | 80.67 | 66.49 | 22.49 | 53.39 | 83.96 | 127.9 |
| | AMSR2_A | 85.04 | 86.84 | 39.1 | 98.53 | 107.01 | 107.72 | 84.72 | 33.53 | 66.73 | 99.12 | 140.03 |
| | AMSR2_D | 79.69 | 88.13 | 37.98 | 88.32 | 103.38 | 101.86 | 82.48 | 31.61 | 65.06 | 98.25 | 137.66 |
| | Multifactor model | 54.79 | 45.46 | 17.82 | 32.54 | 54.67 | 52.41 | 37.48 | 19.71 | 35.67 | 49.63 | 70.92 |





**Table 7 (continued)**

| Parameter | | Elevation (in m) | | | | | | | SCDs (in Days) | | | |
|---|---|---|---|---|---|---|---|---|---|---|---|---|
| Parameter range | | 2000-2500 | 2500-3000 | 3000-3500 | 3500-4000 | 4000-4500 | 4500-5000 | >5000 | 0 - 30 | 30 - 60 | 60 - 90 | > 90 |
| **Upper Himalayan Zone** | In situ SD | - | - | - | 15.38 | 67.8 | 80.33 | 130.94 | 52.79 | 78.2 | 108.24 | 167.86 |
| | AMSR2_A | - | - | - | 17.71 | 77.6 | 104.97 | 188.67 | 97.63 | 130.2 | 157.39 | 205.89 |
| | AMSR2_D | - | - | - | 19.5 | 69.38 | 104.42 | 182.12 | 92.54 | 123.51 | 158.46 | 204.15 |
| | Multifactor model | - | - | - | 11.73 | 34.22 | 55.45 | 126.13 | 83.4 | 89.9 | 108.71 | 122.22 |