# Peer review of "Passive Microwave Remote Sensing based High Resolution Snow Depth Mapping for Western Himalayan Zones using Multifactor Modelling Approach"

_The Cryosphere, 2023_

## Author Comment (AC1)

**Response to editor:**

The authors thank the editor, and highly appreciate the efforts made by the editorial team in handling the manuscript. The overall review comments indicate positive feedback with minor revision of the manuscript and for further consideration. The authors have taken sincere efforts to address each concern raised by the reviewer to improve the manuscript. The comments raised by the reviewer and the response from authors are added below.

**Response to reviewer 1:**

The authors sincerely thank reviewer-1 for his valuable efforts in reviewing our manuscript titled "Passive Microwave Remote Sensing based High Resolution Snow Depth Mapping for Western Himalayan Zones using Multifactor Modelling Approach". The suggestions and feedback shared by the reviewer are highly helpful in enhancing the manuscript. The response to the queries and suggestions provided by the reviewer are attached in the response document below in the point-by-point manner. Kindly note that reviewer comments are in black colour font, response from the author is in blue colour font, whereas the changes made in the manuscript are highlighted in blue colour italic font.

**Reviewer feedback:** Authors have developed a region-wise multifactor model to map the snow depth at 500m spatial resolution in the western Himalayas (particularly three lower, middle and upper zones). This study is essential regarding hydrology, climatology and other natural hazard perspectives and is recommended for consideration.

**Author response:** We thank the reviewer for his kind remarks and overall positive feedback about the manuscript.

**Reviewer comment 1(a):** Under the abstract, the authors have written about the very limited studies that were conducted on snow depth estimation using the passive microwave dataset. I suggest adding more information for more clarity.

**Author Response:** Thank you for suggesting this. The statement mentioning the limited number of studies i.e., 'However, only a limited number of PMW SD estimation studies are carried out for WH till date' is revised with additional details for providing more clarity to the readers. Further it must be noted that all necessary studies in a comprehensive manner are already included in the introduction of the manuscript (kindly refer the information present from Line-80 to Line-125 in the introduction section of the manuscript).

*'However, fewer PMW SD estimation studies are carried out for WH till date, which are mainly confined to small subregions of WH.'*

**Reviewer comment 1(b):** I also advised them to shorten the abstract by removing the information about the LHZ, MHZ, and UHZ (this is obvious information).

**Author Response:** The authors sincerely acknowledge the suggestion given by the reviewer that the statement in the abstract mentioning about the division of WH into LHZ, MHZ and UHZ is obvious information. However, the authors opine it would be difficult to make the reader understand the results part of the abstract without introducing LHZ, MHZ, and UHZ. Additionally, it can be confusing to few of the readers whose region of interest is not Himalayan if LHZ, MHZ, and UHZ are not introduced prior. Therefore, authors believe it is necessary to state that WH is divided into three zones i.e., LHZ, MHZ, and UHZ for which models are built separately and request the reviewer to consider it.

**Reviewer comment 1(c):** Are you sure about the different regression approaches developed in this study? Recheck this statement.

**Author Response:** We thank the reviewer for pointing this out. The authors agree that the statement – 'Different regression approaches (i.e., linear, logarithmic, reciprocal, and power) are developed and evaluated ….' is not very clear as the study focus is standard regression approaches (i.e., linear, logarithmic, reciprocal, and power) to develop the snow depth models using different input variables. Therefore, the statement is revised as suggested by the reviewer.

*'Different regression approaches (i.e., linear, logarithmic, reciprocal, and power) are used to develop snow depth models, which are evaluated further to find if any of these models can address the heterogeneous association between SD observations and PMW TB.'*

**Reviewer comment 1(d):** Mention the full form of AMSR2.

**Author Response:** The full form of AMSR2 i.e., Advanced Microwave Scanning Radiometer 2 is already present in both abstract as well as in manuscript.

**Comment 1(e):** Results need to be mentioned in a precise manner.

**Authors response:** Authors welcome the suggestion shared regarding the results mentioned in the abstract. The information pertaining to results is now revised as given below.

*'Based on a detailed analysis of the results, it is observed from the analysis that power regression SD model has improved accuracy in all WH zones with the less Root Mean Square Error (RMSE) in MHZ (i.e., 27.21 cm) compared to LHZ (32.87 cm) and UHZ (42.81 cm). Spatial distribution of model derived SD is highly affected by SCDs, terrain parameters, geolocation parameters and have better SD estimates compared to regional and global products in all zones. Overall results indicate that the proposed multifactor SD models have achieved higher accuracy in deep snowpack (i.e., SD >25 cm) of WH compared to previously developed SD models.'*

**Reviewer comment 2(a):** Under the introduction part, the authors have covered all the aspects overall. But proofreading is required in some sentences such as their approach (L105), their models are developed (L110) etc.

**Author Response:** The authors have rephrased the lines suggested by the reviewer. Additionally, proof reading is also carried out to see for any other such occurrences.

L105 is revised as below:

*'Das and Sarwade (2008) used 18.7 GHz and 36.5 GHz horizontally polarized data from AMSR-E and modified the coefficients of Chang et al. (1987)'s model to suit the Indian Himalaya. The modified model has shown a mean absolute error (MAE) of 20.34 cm in SD estimates but failed to estimate SD above 60 cm. '*

L110 is revised as below:

*'In another study, Singh et al. (2015) developed PMW SD models for the Dhundi and Patseo regions of Himalaya using data from ground-borne radiometers and in-situ observations. However, SD models are developed using observations collected from only two field surveys, evaluated using a single day observation of AMSR-E TB data, and not tested spatiotemporally.'*

**Reviewer comment 2(b):** Revision is required for the second objective (L135) due to some grammatical issue. Under the third objective, variables could be mentioned.

**Author Response:** As advised by the reviewer, the second objective has been rephrased and included in the manuscript.

*'Comparison and evaluation of the proposed multifactor model(s), previous SD models and AMSR2 SD products in different WH zones.'*

Also, in the objective variables are now mentioned and included in the manuscript as follows:

*'Analysis of multifactor SD model's retrieval accuracy with respect to the selected auxiliary variables (such as elevation, slope, land cover types, and SCDs).'*

**Reviewer comment 3(a):** Under the study area, many terms are repetitively defined such as LHZ, MHZ, and UHZ (165) as you have already explained under L40. Underline 165 Upper Himalayan Zone is incorrectly abbreviated as MHZ.

**Author Response:** The authors thank the reviewer for rightly pointing this out. Now, authors have corrected the manuscript as advised. The statements mentioning the presence of three zones are now removed to avoid repetition (similar information as given in L40). Further, the line (L165) in the manuscript is removed in the revised manuscript. The updated study area information in the revised manuscript is as follows from line 165 to 170.

*'In this study, three WH zones i.e., LHZ, MHZ, and UHZ defined based on the historical local meteorological and avalanche occurrence data (Sharma and Ganju, 2000) are used for developing multifactor SD models. The geomorphic and climate characteristics of these zones are given in Table 2.'*

**Reviewer comment 3(b):** Under the methodology part, the flowchart is well-defined but the five steps need to be mentioned in the flowchart as explained in the subsections. The flowchart must be the stepwise reflection of the subsection (3.1 to 3.5).

**Author Response:** Based on the reviewer suggestions, the flowchart is now updated as follows in the revised manuscript.

[Figure]

Figure. 2. Flowchart representing the methodology

**Reviewer comment 4:** Results and discussion are well explained. The challenges are also defined in the discussion part. However, could you please also highlight any scope of the advanced machine learning or deep learning approach in the snow depth estimation? (Some

of the previous studies also involved the neural network/deep learning approach in snow depth estimation). I think this point may increase the interest of the readers.

**Author Response:** We thank the reviewer for suggesting this. In this study, the proposed multifactor SD model has an advantage that it uses a specific equation to predict the complex, non-linear relationship between SD and independent multifactor. The SD predicted using the proposed model(s), on the other hand, has a substantial error. The Machine learning algorithms can help get around some of these problems. Different machine learning approaches are widely used for SD estimation in different studies for example artificial neural networks (Tedesco et al., 2004) over Finland, support vector machines (Liang et al., 2015) over Northern China, and Extremely randomized trees (Tanniru and Ramsankaran, 2023) over Alaska. However, specific information related to machine learning approaches and their potential is not in the scope of the current manuscript. Therefore, authors have mentioned and updated some information related to machine learning as a possible scope for improvement in the manuscript under the conclusions and summary section. The revised information is as follows.

*'Recently, different machine learning models are extensively used for modelling SD in many studies (Tedesco et al., 2004, Liang et al 2016, Tanniru and Ramsankaran, 2023). The potential of such machine learning approaches can be investigated for improving the SD estimation.'*

Tedesco, M., Pulliainen, J., Takala, M., Hallikainen, M., and Pampaloni, P.: Artificial neural network-based techniques for the retrieval of SWE and snow depth from SSM/I data, Remote Sens Environ, 90, 76–85, https://doi.org/https://doi.org/10.1016/j.rse.2003.12.002, 2004.

Liang, Jiayong, et al. "Improved snow depth retrieval by integrating microwave brightness temperature and visible/infrared reflectance." *Remote Sensing of Environment* 156 (2015): 500-509.

S. Tanniru and R. Ramsankaran, "Machine Learning-Based Estimation of High-Resolution Snow Depth in Alaska Using Passive Microwave Remote Sensing Data," in *IEEE Journal of Selected Topics in Applied Earth Observations and Remote Sensing*, vol. 16, pp. 6007-6025, 2023, doi: 10.1109/JSTARS.2023.3287410.

**Reviewer comment 5:** The conclusion part needs to be specific. However, it has been observed that many contents of the conclusion don't make any significant impact like the WH region is divided into three zones, i.e., LHZ, MHZ, and UHZ. So, such types of lines could be removed.

**Author Response:** We thank the reviewer for suggesting this. The conclusions section is revised by as suggested by the reviewer.

*'The contrasting climate and snow conditions prevailing in WH zones present new challenges in accurate SD retrievals using PMW remote sensing. The limited access to in-situ SD data, rugged topography, and inclement weather resulted in fewer SD studies over the WH region. In the*

*mountainous region, the topography parameters, i.e., elevation and slope, affect the snow precipitation and its persistence.*

*In this study, different regression approaches (i.e., linear, logarithmic, reciprocal, and power) are used for developing the multifactor SD models using multifrequency AMSR2 TB observations and auxiliary parameters (such as terrain (elevation, slope), location, SCD, etc.,) to estimate SD at 500 m spatial resolution in each WH zone. The overall results indicate power regression performed better compared to other tested approaches in all zones. Further, the results of the multifactor model from power regression are evaluated by comparing the SD estimates with ground SD, other SD products, and PMW models. The results indicate under deep snow (>25 cm) conditions the developed multifactor model has shown higher accuracy compared to the AMSR2 operational SD product and other SD models. However, the accuracy of SD from the multifactor model is affected by variations in auxiliary parameters such as SCD, elevation, etc. With an increase in SCD, the SD increased in each WH zone. Additionally, the RMSE error associated with SD is also increased alongside SCD and SD in each WH zone. The MHZ has stable snow conditions with relatively less thick snowpack. Therefore, the multifactor SD model in this region has shown improved accuracy for a given SD class compared to other WH zones. Overall, the proposed multifactor SD models for WH zones have demonstrated substantial improvement in estimating SD compared to the operational AMSR2 SD product, heritage SD model, i.e., Chang's model, and previous models developed within WH zones.*

*Though multifactor SD model has outperformed other tested models and products, there is still scope for improving PMW SD estimates in WH. The developed model(s) have shown poor performance compared to AMSR2 products when SD <25 cm. This can be possibly attributed to wet snow conditions prevailing in the early winter, i.e., when SD will be shallow. Further, the inclusion of snowpack characteristics such as snow grain size, wetness, density data during the model development can improve the accuracy of SD estimates. The available in-situ SD observations are very limited considering the high spatiotemporal variability of SD in this region. Therefore, there is an immediate need of expanding the in-situ network of monitoring stations, and field-based studies to determine the first-hand knowledge of snowpack information in WH region. Recently, different machine learning models are extensively used for modelling SD in many studies (Tedesco et al., 2004, Liang et al 2016, Tanniru and Ramsankaran, 2023). The potential of such machine learning approaches can be investigated for improving the SD estimation.'*

**Reviewer comment 6:** Overall, many aspects have been disclosed in this article and recommended for further consideration.

**Authors response:** We greatly thank the reviewer for the kind remarks on the manuscript.

---

## Author Comment (AC2)

**Response to the comments of reviewer 2 (RC3)**

The authors sincerely thank reviewer-2 for his valuable efforts in reviewing our manuscript titled "Passive Microwave Remote Sensing based High Resolution Snow Depth Mapping for Western Himalayan Zones using Multifactor Modelling Approach". The suggestions and feedback shared by the reviewer are highly helpful in enhancing the manuscript. The response to the queries and suggestions provided by the reviewer are attached in the response document below in the point-by-point manner. Kindly note that reviewer comments are in black colour font, response from the author is in blue colour font, whereas the changes made in the manuscript are highlighted in blue colour italic font.

**Reviewer feedback:** The authors present a novel technique for modelling snow depths using passive microwave observations in the Western Himalayan region. Their multiparameter approach is compelling and appropriate for the study area and remote sensing datasets utilized. The created multifactor model provides spatially distributed estimates of snow depth at a 500 m resolution, which are necessary for hydrologic modelling and understanding natural hazard risk in the region. While the author's model shows promising results for the Western Himalayan region, the following points may be addressed to improve the clarity and strength of the analysis.

**Author response:** The authors would like to sincerely thank the reviewer for his generous feedback about the manuscript.

**Notes:**

**Reviewer comment 1:** I recommend additional proofreading for the manuscript. Articles are used incorrectly or missing in places, and at times grammatical mistakes interfere with the meaning of the text.

**Author response:** The authors sincerely acknowledge the feedback and suggestions given by the reviewer. Additional proofreading of the manuscript is now carried out with help of Grammarly tool. The authors have verified and corrected grammatical mistakes as observed while proofreading, and those pointed by the Grammarly tool. Several sentences are rephrased to ensure that the information is conveyed clearly to the reader.

**Reviewer comment 2:** Acronym use is high in the manuscript and can lead to confusion while reading. Removing some of the lesser-used acronyms would improve.

**Author response:** The authors do agree that many acronyms are present in the manuscript. However, the authors opine that the acronyms present in the manuscript are necessary and used accordingly. Therefore, authors would like to proceed without removing the acronyms.

**Reviewer comment 3:** Repetition is an issue in the manuscript. Some sections of the Results and Discussion start by repeating methods or goals (e.g., lines 338, 354-356, 374-376, etc.). Please reduce repetition to improve the clarity of the text.

**Author response:** Thanks for pointing this out. The authors have done the proofreading of the entire manuscript. The following lines in the reviewed (original) version of the manuscript are removed to reduce the repetition.

Removed lines: L:338-339, L:354-356, L:374-376, L498-502

**Reviewer comment 4:** The Discussion section would be strengthened if the created multifactor model was discussed in more detail. This could include why the authors' model outperformed existing models, transferability to other mountain ranges, and model error. Summaries of the methods and results are overemphasized in this section.

**Author response:** The authors thank the reviewer for providing this valuable suggestion. The discussion section is now revised as suggested by the reviewer by including the details about – model error, why the model has outperformed existing models, and model's transferability to other regions. Following information is added for improving the discussion section.

*"The regression modelling approach attempts to find a better fit by optimizing the loss function i.e., mean error. Over WH region, majority of the observations have SD > 25 cm. Therefore, understandably the model estimates are better in higher SD regions as compared to shallow SD regions."*

*"The topographical parameters in WH play a vital role in affecting the local climate as well as snow distribution. The inherent weakness of PMW TB in capturing deeper snowpack thickness is overcome to certain extent by considering SCD into model development. Thus, the overall improved performance of multifactor model over the previously developed other models and AMSR2 product can be attributed to the consideration of topographical parameters and SCD into the model development. Further, combination of multiple lower and higher frequency TB is considered into the model for capturing both deeper and shallower snowpack thickness. Different factors affecting the performance of the multifactor SD model are discussed in detail in the following section, 5.2."*

*"The developed model has shown improved performance as compared to other tested approaches in the WH region. However, the transferability of the multifactor model to the other regions specifically mountainous regions is uncertain. This is due to the fact that the relationship of SD with topographical conditions and SCD can potentially change in the other regions. The proposed multifactor model coefficients attempt to improve SD estimates as per prevailing snow conditions in WH. Understanding the influence of topographical conditions, and snow persistency, and snow pack dynamics is essential for using the model outside the WH."*

**Specific Comments**

1**:** Line 155: The area of the study region should read "360,866 km$^2$."

**Author response:** The numbering format is adjusted as suggested by the reviewer. Now the revised line is as follows.

*"WH extends between longitudes from 73° 15' E to 79° 45' E, latitudes from 30° 00' N to 39° N and covers an area of 360,866 km$^2$."*

2. Line 193: How is the AMSR2 snow depth product created? A citation would be helpful.

**Author response:** Thanks for suggesting this. Reviewer #3 also suggested the same. The details regarding the AMSR2 SD product development along with the citation are now added into a new section 2.4 in the revised manuscript.

*"In this study, the AMSR2 SD products have been downloaded from the website (https://gportal.jaxa.jp) during the snow season (October to March) from 2012 to 2019. The SD products corresponding to ascending (13:30 ± 15 min) and descending (01:30 ± 15 min) pass have been used for comparison with the multifactor model SD estimates. The standard AMSR2 SD algorithm primarily uses the daily 10, 18, 23, 36, and 89 GHz frequencies brightness temperature data and the surface physical temperature (T) data. In the development of the AMSR2 SD algorithm (Kelly 2009), the following steps and conditions have been considered.*

*Step 1- Isolate wet and dry snow/no-snow-covered regions: If dry snow is present in any region, it will satisfy the conditions (1) and (2)(move to step 2); otherwise, there is no snow-covered region, or only wet snow is present*

$$Tb_{36H} < 245K \tag{1}$$
$$Tb_{36V} < 255K \tag{2}$$

*Step 2- Isolate moderate/deep and shallow snow-covered areas: If moderate/deep snow is present, it will satisfy the conditions (3) and (4) (move to step 4) (Derksen 2008); otherwise, shallow snow is present or no snow-covered area (move to step 3)*

$$Tb_{10H} - Tb_{36H} > 0K \tag{3}$$
$$Tb_{10V} - Tb_{36V} > 0K \tag{4}$$

*Step 3- Identify a shallow snow-covered area. If it satisfies conditions in (5), then shallow snow is present, and a flag of 5.0 cm is set for the SD; otherwise, no snow is present*

$$Tb_{89V} < 255K, \ Tb_{89H} < 265K, Tb_{23V} > Tb_{89V}, Tb_{23H} > Tb_{89H} \ and \ T < 267K \tag{5}$$

*Step 4: Estimation of moderate to deep SD using Equation (6)*

$$SD = \left[\frac{1}{log_{10}(Tb_{36V} - Tb_{36H})X(Tb_{10V} - Tb_{36V})}\right] \qquad (6)$$
$$+ \left[\frac{1}{log_{10}(Tb_{18V} - Tb_{18H})X(Tb_{10V} - Tb_{18V})}\right]$$

*The developed SD algorithm by Kelly, 2009 was tested using World Meterological Organization (WMO) collected SD measurements from 242 and 254 sites around world during the 2002-2003 and 2003-2004 winter season, respectively. In this only non-mountain stations with at least 30 days of measured snow were used in the comparison. In the recent study conduct over the mountainous terrain of Northern Xinjiang Region, China by the Zhang et al. (2017) the AMSR2 SD products were compared with ground collected SD data. They observed RMSE of 18.5 cm (in AMSR2_A) and 23.4 cm (in AMSR2_D) up to 30 cm of ground SD. However, AMSR2 SD products have not been evaluated for Indian Western Himalayan regions till date."*

3. Line 194: The link did not work for me.

**Author response:** The working link of the JAXA data archive portal i.e., https://gportal.jaxa.jp is updated in the revised manuscript.

4. Line 210: "SCD" is not defined in the main body of the manuscript.

**Author response:** Thanks for pointing this out. Reviewer #3 (RC4) suggested some changes with regard to SCD. The revised manuscript defines the SCD at its first instance.

*"Snow cover duration (SCD) depicts the number of consecutive days snow cover is present for a given pixel."*

5. Line 214: More connection is needed to the studies from Sharma et al. (2014) and Singh et al. (2018). Were methods from these publications followed to estimate snow cover days?

**Author response:** The authors thank the reviewers for suggesting this. The studies by Sharma et al. (2014) and Singh et al. (2018) have calculated SCD, however their research interest is different. Their prime focus is on understanding the spatiotemporal variation of the snow cover, and SCD trends over the WH region. It is important to note that the exact methodology used by Sharma et al. (2014) is not described in the publication. Authors have independently calculated the SCD from the MODIS cloud free snow cover product for each water year. The details regarding how the SCD is calculated are added. Reviewer #3 (RC4) also made some suggestions with regard to SCD. Therefore, the connection with regard to Sharma et al. (2014) and Singh et al. (2018), and suggestions from reviewer 3 are now updated in the revised manuscript.

*"In WH, snow cover area (SCA)/snow cover pixels vary during different months of the year due to change in snowfall and snow ablation pattern. Least SCA has been observed during the month of August/September and maximum SCA was observed during the month of February/March. Snow cover duration (SCD) depicts the number of consecutive days snow cover is present for a given pixel. It provides information regarding the persistence of snowpack and is useful in improving PMW SD estimates (Singh et al., 2016; Wang et al., 2019; Dai et al., 2018). In this study, daily could-free MODIS snow cover product (i.e., M\*D10A1GL06) generated for high-mountain Asia (Muhammad and Thapa, 2020) at 500 m spatial resolution (https://doi.org/10.1594/PANGAEA.918198) has been used to generate SCD product for the study area during the data period. Previously, Sharma et al. (2014) and Singh et al. (2018) have generated and evaluated the SCD maps for snow-covered Indian WH. These studies (Sharma et al., 2014, Singh et al., 2018) revealed a higher average monthly SCD (>80%) in high-altitude regions. These studies' results further emphasize a strong longitudinal and altitudinal dependence on SCD, snow cover accumulation and ablation in WH. Therefore, SCD information can provide valuable insights to improve the SD model. Daily binary snow cover maps prepared from M\*D10A1GL06 are used to identify the snow cover presence for a given pixel. These binary snow cover maps are used for computing the SCD information for each day from October 1st of each year to September 30th of the following year during the study period."*

6. Line 222: Was land cover different in the study period (i.e., 2012-2018) from the 2019 composite? How might this impact results?

**Author response:** There is no change in the land cover classes over the selected stations during the study period. Therefore, it will not impact the results. However, authors agree that landcover changes elsewhere in the WH region during the study period. Therefore, while generating the daily snow depth products recently available landcover dataset for that time shall be used.

7. Line 235: What was the impact of resampling the AMSR2 brightness temperature observations? How do results change if the spatial resolution is increased or decreased?

**Author response:** The reviewer concern about the spatial resolution is duly noted by the authors. It is an interesting idea to test with different resolutions. The change in resolution may have an impact on the snow depth estimation results, however it is beyond the scope of what authors have proposed in this work. However, the authors would like to point that, the brightness temperature data used in this work are actually resampled instead of downscaling. Further, the main motivation behind this work is to develop the snow depth at high resolution i.e., 500 m. The ideas suggested by the reviewer however are already under consideration by the authors. Downscaling the brightness temperature to different resolutions then resampling the other datasets can be tested in the future works for investigating if it can improve the model estimates. Considering the amount of work involved in implementing this

is considered as future scope for improvement. This information is now included into the manuscript as potential scope for improving this research work.

*"The brightness temperature datasets used in this work are resampled to 500 m. Instead of resampling, downscaling the TB can be tested for further improvement of model. It is also worthwhile to investigate how downscaling the TB to different resolutions will impact the model performance."*

8. Line 236: What software was used to resample images, reproject images, and calculate brightness temperature difference? Does any other processing need to be done to AMSR2 brightness temperature data or is it done already?

**Author response:** The reprojection of images is carried out using format conversion tool. Resampling is performed with help of ArcGIS software. Python programming is used for retrieval of TB and calculation of BTD. No additional processing is carried out. The detailed information is incorporated into the revised manuscript.

*"The brightness temperature and SD datasets downloaded from JAXA portal have northern hemisphere polar stereographic coordinate system and are present in the HDF5 format. These are reprojected to WGS 1984 coordinate system and are converted to tiff format with help of format conversion tool developed by the JAXA. Following that ArcGIS software is used for resampling the BT imagery to 500 m. No additional processing is carried out in the current work as the brightness temperature dataset acquired from JAXA are level-3 product. However, the brightness temperature is corrected for forest cover fraction in locations where vegetation is present as per the foster model. The brightness temperature from each image for all stations is then retrieved programmatically using python. The extracted TB data is used for calculating the BTD."*

9. Line 380: Was bias in the modelled estimates considered? I am curious if the models consistently over or underestimated snow depths.

**Author response:** Thank you for suggesting this. The bias is estimated by the authors in this work is now included in the discussion section of the revised manuscript.

*"The proposed model has an overall positive bias with overestimated SD values for lesser SD, and underestimation in the case of higher SD observations. The bias for LHZ, MHZ and UHZ for the proposed model is 4.5 cm, 2 cm and 6.3 cm respectively. Whereas the bias for legacy model, and other regional model is considerably higher with substantial overestimates in the lower depth values and underestimates in higher depth regions. Further, it must be emphasized that these models have very poor correlation with the in-situ snow depth and the*

*SD estimates mainly confined in a range irrespective of the magnitude of the ground snow depth observation values."*

10. Line 402: It is unclear to me why results from the MHZ refer to Figure 6 when the caption for Figure 6 states the reported regions are in the LHZ.

**Author response:** Thanks for pointing this, it is a typographical mistake. The figure caption is revised.

11. Line 410: It is unclear to me why results from the UHZ refer to Figure 6 when the caption for Figure 6 states the reported regions are in the LHZ.

**Author response:** Thanks for pointing this, it is a typographical mistake. The figure caption is revised.

12. Line 492: Figure 10(d) does not exist.

**Author response:** Thanks for pointing this, this is typographical mistake. It shall be Figure 8(d). Now it is corrected in the revised manuscript.

13. Line 532: Correlations of less than 0.5 are generally considered weak or moderate.

**Author response:** The authors agree with the reviewer. The statement is revised to include the word moderate in the place of strong.

14. Line 534: How did local incidence angles impact the accuracy of snow depth estimates?

**Author response:** PMW brightness temperature can be affected when nominal incidence angle is above $50^{\circ}$ and the terrain slope exceeds $20^{\circ}$ (Che et al., 2011). Considerable amount of area in WH exhibits a slope of above $20^{\circ}$. However, this effect on snow depth model can be minimized when Brightness Temperature Difference (BTD) is used for the model development. Further, the study (Che et al., 2011) demonstrated that local incidence angle has lesser sensitivity on SD retrievals. Therefore, this will have very less impact on the SD retrievals of multifactor model as BTD are used instead of single channel brightness temperature.

Che, T., Dai, L., Wang, J., Zhao, K., & Liu, Q. (2012). Estimation of snow depth and snow water equivalent distribution using airborne microwave radiometry in the Binggou Watershed, the upper reaches of the Heihe River basin. *International Journal of Applied Earth Observation and Geoinformation, 17,* 23-32

15. Figure 1, Figure 4: The acronyms "J&K" and "HP" are not defined in the figure caption.

**Author response:** The acronyms are now defined in the captions of figure1, and figure4.

16. Figure 5: Units for standard deviation are missing. Standardizing the standard deviation may improve the interpretability of the figure.

**Author response:** The units for standard deviation are now updated in the revised figure 5. The standardization of standard deviation is not very common in many of the papers that are published as authors have observed. The origin plotting tool which authors have employed for making the taylor diagram natively does not allow to standardize the standard deviation. Though there are some other python packages and tools available for doing this, the authors have taken feedback from few other members. After consultation, the authors opined that the figure with the standardized standard deviation can also be equally difficult to interpret for some of the community unlike what the reviewer has mentioned. Therefore, authors suggest that it would be simpler to proceed with the same figure 5.

[Figure]

*Figure. 5. Taylor diagram for the evaluation of multifactor SD models during 2017-2019 for (a) the LHZ, (b) the MHZ, and (c) the UHZ.*

17. Figure 6: Where are the Pir-Panjal, Greater Himalayan, and Karakoram regions? Perhaps these regions could be marked in Figure 1.

**Author response:** The figure 1 is updated to include the information pointed by the reviewer.

[Figure]

*Figure. 1. (a) Elevation variability of WH zones (i.e., LHZ: Lower Himalayan Zone; MHZ: Middle Himalayan Zone; UHZ: Upper Himalayan Zone) and DGRE observatories distribution. (Note: J&K, H.P. are Jammu and Kashmir, Himachal Pradesh respectively)*

18. Figure 7: The scale for subplots b, c, and d should be the same for comparison, especially at depths of 0 cm and for regions with missing data.

**Author response:** Thanks for suggesting this. The figure 7 is revised to ensure that that scale is consistent for all the subplots.

[Figure]

*Figure. 7: Spatial map of SD variation on 3rd Feb 2019. (a) MODIS SCA, (b) AMSR2_A SD product map at 10 km, (c) AMSR2_D SD product map at 10 km, and (d) multifactor models SD map at 500 m.*

19. Figure 8: RMSE may be better shown as scatter plots so variations in error between elevation, slope, land cover types, and snow cover days are more apparent. As depicted, these patterns are hard to assess visually considering the large study area, scale, and overlap between the stations. Further, this figure is discussed in terms of the lower, middle, and upper Himalayan zones (lines 465-470); however, only state borders are drawn on the map (rather than zone boundaries).

**Author response:** Figure 8 is updated with boundaries of zones. Additionally figure. 9 (i.e., scatter plots) is prepared as suggested by the reviewer. However, the authors believe that it would be difficult to discern any strong pattern even from the scatter plots. This is primarily due to the lesser number of stations and large variation in the error for stations at given elevations. Furthermore, the some of the stations in lower, middle and upper Himalaya have similar elevations despite having different climate and snow conditions. This would add to the complexity in interpreting the accuracy through scatter plots. Further, when it comes to slope, landcover there is not clear pattern in the scatter plot. With regard to SCDs, though no trend is observed it can be seen higher RMSE is present for stations having higher SCD.

[Figure]

*Figure 8. Spatial distribution of RMSE of multifactor SD model for varying (a) elevation, (b) slope, (c) land cover types, and SCDs along the 43 ground stations*

[Figure]

*Figure 9. Scatter plots for RMSE of multifactor SD model for varying (a) elevation, (b) slope, (c) land cover types, and SCDs of the 43 ground stations*

20. Table 1: The first two links in the table did not work for me. Full citations and access dates would also be helpful.

**Author response:** The two links in the table are updated now with the following respectively.

1. https://gportal.jaxa.jp (last accessed on: 26/ (last accessed on: 26/11/2023)
2. https://lpdaac.usgs.gov/products/mcd12q1v006/ (last accessed on: 26/11/2023)

21. Table 6: Was the number of observations consistent between snow depth classes? On line 455 it is stated that only four of the 43 in-situ stations have mean snow depths of less than 25 cm. If there are fewer observations within this class it may influence the resulting error.

**Author response:** The authors would like to inform the reviewer that the number of samples considered for calculating accuracy metrics in each snow depth class are different. As mentioned in the manuscript, there are only four stations which have a mean snow depth of less than 25 cm. However, it is important to note that all stations have observations with SD < 25 cm, though mean is higher than 25 cm. Approximately 20% of the in-situ SD observations in WH have SD <25 cm and represent sufficient number of samples in each WH zone. A figure representing the distribution of in-situ SD observations is attached here for reviewer's consideration. The metrics for each SD class is calculated by matching the estimates from each SD product with in-situ observations. Therefore, the authors believe it is acceptable to use the way samples are used for evaluation of model estimates.

[Figure]

*Figure: Cumulative distribution of in-situ SD over WH*

22. Table 7: The font for 'Lower Himalayan Zone,' 'Middle Himalayan Zone,' etc. should face the same direction as model names and RMSE values.

**Author response:** The font facing direction is adjusted accordingly as pointed out by the reviewer.

---

## Author Comment (AC3)

The authors sincerely thank reviewer-3 for his valuable efforts in reviewing our manuscript titled "Passive Microwave Remote Sensing based High Resolution Snow Depth Mapping for Western Himalayan Zones using Multifactor Modelling Approach". The suggestions and feedback shared by the reviewer are highly helpful in enhancing the manuscript. The response to the queries and suggestions provided by the reviewer are attached in the response document below in the point-by-point manner. Kindly note that reviewer comments are in black colour font, response from the author is in blue colour font, whereas the changes made in the manuscript are highlighted in blue colour italic font.

**Reviewer feedback:** This is a worthwhile and interesting study that explores new snow depth retrieval methodologies using passive microwave observations and associated datasets such as terrain and MODIS snow cover. A particular strength is the focus on the Western Himalaya and the use of a substantial dataset of in-situ snow depth measurements for training and validating the snow depth retrievals. An interesting aspect of the study is the way the high-resolution datasets (e.g., MODIS and terrain) are used to localise the 10 km passive microwave dataset down to 500 m. While the study could have used a more sophisticated machine learning approach, it instead uses multiple-parameter regression models, including linear, power, logarithmic and so on. This is actually a very positive aspect of the study as the resulting simple models are clear and can easily be re-used by other investigators, both for retrievals and as a simple way of assessing sensitivity of snow depth to the various different parameters.

The work is clearly written and mostly well-presented, although some copy editing would be required in places. The main issue to be addressed is the use of the AMSR2 official snow depth product as a reference against which to assess the performance of the new algorithm. As is shown in figure 7, the official algorithm provides zero snow depth over most of the Western Himalaya at a moment when MODIS suggests the area is almost completely snow covered. The areas where AMSR2 does produce non-zero snow depths appear to be topographically linked - for example lower or flatter terrain areas. This suggests the AMSR2 algorithm is not working adequately, so it is a poor reference against which to compare. I wonder if it is possible that some bad quality flags have been set for this product, even if the data is not officially flagged as missing. The most problematic aspect of the use of the AMSR2 snow depth is that thanks to the zero values, it has apparently better performance in low snow depth conditions, in RMS error terms, than the new multi-factor retrieval. This is almost certainly a spurious result (it is surely better to produce snow where it lies, even if too deep, than to "game" the results by setting almost all depths to zero). The comparison to AMSR2 needs to be de-emphasised within the results, and ideally a more adequate reference should be chosen, such as one of the legacy algorithms investigated elsewhere. Given this one main issue and a number of smaller but important points, mainly relating to clarifying the work so that it can be re-used other scientists, I recommend major revisions.

**Author response:** The authors thank the reviewer for the constructive feedback for improving our manuscript. The authors have done additional proofreading of the manuscript with the help of Grammarly tool to improve the manuscript.

The authors partially agree with the reviewer's opinion that AMSR 2 is a poor reference for comparing the performance of the multifactor model. AMSR2 SD and GlobSnow SD products are the operational PMW SD products available at the time of this manuscript's preparation. GlobSnow product completely masks the WH region and provides no SD information. The authors' also have similar observations, i.e., the AMSR2 SD product having better performance in shallow snow depth conditions is spurious because most of the observations given by AMSR2 had SD<60 cm. The authors' intention in comparing the AMSR2 SD product with multifactor model is to demonstrate (i) the quality of the AMSR2 product over WH (which is not presented previously by other researchers) and (ii) how the developed model is performing against the operational products. This has been already mentioned in the L 135 of the manuscript. Apart from comparison with the AMSR2 SD product, the authors have also compared one legacy model, i.e., Chang's model, and two regional models in this study, as given in different sections of the manuscript. Further, it is essential to note that the comparison against the AMSR2 product and other empirical models is only part of the evaluation. The model is primarily evaluated by comparing it with the in-situ SD observations collected from the snow observatories during the testing period, i.e., 2017-18 to 2018-19.

**Response to major comments:**

**Reviewer comment 1**: The AMSR2 snow depth product needs to be described in more detail in the relevant methodology section, e.g., an overview of how the AMSR2 product is derived, whether there are any quality flags or areas where the product is known to perform poorly.

**Author Response:** We are thankful to the reviewer for the valuable suggestions. The detailed description about the AMSR2 SD product, product generation, and its performance is provided as a new subsection (section 2.4) under the study area and datasets section.

*"In this study, the AMSR2 SD products have been downloaded from the (https://gportal.jaxa.jp) during the snow season (October to March) from 2012 to 2019. The SD products corresponding to ascending (13:30 ± 15 min) and descending (01:30 ± 15 min) pass have been used for comparison with multifactor model estimates. The standard AMSR2 SD algorithm primarily uses the daily 10, 18, 23, 36, and 89 GHz frequencies brightness temperature data and the surface physical temperature (T) data. In the development of the SD algorithm (Kelly 2009), the following steps and conditions have been considered.*

*Step 1- Isolate wet and dry snow/no-snow-covered regions: If dry snow is present in any region, it will satisfy the conditions (1) and (2) (move to step 2); otherwise, there is no snow-covered region, or only wet snow is present*

$$Tb_{36H} < 245K \qquad (1)$$
$$Tb_{36V} < 255K \qquad (2)$$

*Step 2- Isolate moderate/deep and shallow snow-covered areas: If moderate/deep snow is present, it will satisfy the conditions (3) and (4) (move to step 4) (Derksen 2008); otherwise, shallow snow is present or no snow-covered area (move to step 3)*

$$Tb_{10H} - Tb_{36H} > 0K \qquad (3)$$
$$Tb_{10V} - Tb_{36V} > 0K \qquad (4)$$

*Step 3- Identify a shallow snow-covered area: If it satisfies conditions in (5), then shallow snow is present, and a flag of 5.0 cm is set for the SD; otherwise, no snow is present*

$$Tb_{89V} < 255K, \; Tb_{89H} < 265K, Tb_{23V} > Tb_{89V,} \; Tb_{23H} > Tb_{89H} \; and \; T < 267K \qquad (5)$$

*Step 4: Estimation of moderate to deep SD using Equation (6)*

$$SD = \left[ \frac{1}{log_{10}(Tb_{36V} - Tb_{36H})X(\,Tb_{10V} - Tb_{36V})} \right] \qquad (6)$$
$$+ \left[ \frac{1}{log_{10}(Tb_{18V} - Tb_{18H})X(\,Tb_{10V} - Tb_{18V})} \right]$$

*The developed SD algorithm by Kelly 2009 was tested using World Meterological Organization (WMO) collected SD measurements from 242 and 254 sites around world during the 2002-2003 and 2003-2004 winter season, respectively. In this only non-mountain stations with at least 30 days of measured snow were used in the comparison. In the recent study conduct over the mountainous terrain of Northern Xinjiang Region, China by the Zhang et al. (2017) the AMSR2 SD products were compared with ground collected SD data. They observed RMSE of 18.5 cm (in AMSR2_A) and 23.4 cm (in AMSR2_D) upto 30 cm of ground SD. However, AMSR2 SD products have not been evaluated for Indian Western Himalayan regions till date.”*

**Reviewer comment 2**: AMSR2 instrument should have a citation and there should be a DOI for the data (as for other datasets used).

**Author Response:** As suggested citation for AMSR2 instrument is updated, and DOI for the other datasets are added wherever available in revised manuscript.

*"AMSR2 is a PMW sensor onboard the Japanese Aerospace Exploration Agency (JAXA)'s Global Change Observation Mission 1ˢᵗ - Water (GCOM-W1) SHIZUKU, launched in May 2012 (Imaoka et al., 2012)."*

DOI for MODIS landcover data: https://doi.org/10.5067/MODIS/MCD12Q1.061

*Imaoka, K., Maeda, T., Kachi, M., Kasahara, M., Ito, N., & Nakagawa, K. (2012, November). Status of AMSR2 instrument on GCOM-W1. In Earth observing missions and sensors: Development, implementation, and characterization II (Vol. 8528, pp. 201-206). SPIE*

**Reviewer comment 3**: The multi factor model targets dry snow conditions and hence the study is performed only in the winter period. It would be helpful to mention this in the abstract and conclusions since this appears to be the main limitation on the validity of the model, other than the geographical specificity.

**Author Response:** The authors are thankful to reviewer for the valuable suggestion and totally agree with reviewer's observation; therefore, in the revised manuscript changes are made in the abstract and conclusion section as suggested by the reviewer.

Abstract section - *"Multifrequency brightness temperature (TB) observations from Advanced Microwave Scanning Radiometer 2 (AMSR2), SCDs data, terrain parameters (i.e., elevation, slope and ruggedness), geolocation for the winter period (October to March) during 2012-13 to 2016-17 are used for developing the SD models for dry snow conditions."*

Conclusions section - *"The multifactor model is applicable only to dry snow conditions. However, in WH even during the peak winter substantial area is covered by wet snow. This constrains the utility of multifactor model for these regions."*

**Reviewer comment 4**: Of the 40-brightness temperature difference (BTD) pairs tested, only 8 are retained. The paper describes these 8 BTDs, but not the other 32. It is important to describe also which BTD pairs were rejected, to indicate what does not predict snow depth.

**Author Response:** As suggested, along with the 8 BTD pairs that are retained, the 32 rejected BTD pairs have been added into the Table of the revised manuscript.

**Table 4. BTD SD model (with descending observations) relation with SD and evaluation using LOOCV method**

| | Sr. No. | Independent Variable (x) | Linear Regression Model | RMSE (in cm) | R |
|---|---|---|---|---|---|
| **Selected parameters** | 1 | BTD (36H-89V) | y = - 2.24x + 107.05 | 91.63 | 0.39 |
| | 2 | BTD (36V-89V) | y = - 2.16x + 81 | 92.24 | 0.37 |
| | 3 | BTD (10V-23H) | y = 4.12x + 31.05 | 92.45 | 0.35 |
| | 4 | BTD (23H-89V) | y = - 1.78x + 122.17 | 92.46 | 0.36 |
| | 5 | BTD (10V-18V) | y = 7.43x + 52 | 92.58 | 0.25 |
| | 6 | BTD (10H-23H) | y = 4.12x + 56 | 93.78 | 0.20 |
| | 7 | BTD (10H-18H) | y = 5.66x + 58 | 93.47 | 0.21 |
| | 8 | BTD (18H-89V) | y = - 1.61x + 122.34 | 93.92 | 0.24 |
| **Rejected parameters** | 9 | BTD (10H-36H) | y = 0.85x + 70.11 | 102.20 | 0.17 |
| | 10 | BTD (10H-89H) | y = -0.91x + 114.15 | 102.16 | 0.18 |
| | 11 | BTD (10H-18V) | y = 2.84x + 89.85 | 102.20 | 0.17 |
| | 12 | BTD (10H-23V) | y = 3.29x + 78.28 | 102.15 | 0.18 |
| | 13 | BTD (10H-36V) | y = 0.55x + 77.67 | 102.21 | 0.16 |
| | 14 | BTD (10H-89V) | y = -1.15x + 177.36 | 102.14 | 0.19 |
| | 15 | BTD (10V-18H) | y = 5.10x + 30.11 | 102.04 | 0.20 |
| | 16 | BTD (10V-36H) | y = 1.21x + 56.24 | 102.17 | 0.18 |
| | 17 | BTD (10V-89H) | y = -0.66x + 110.16 | 102.19 | 0.17 |
| | 18 | BTD (10V-23V) | y = 4.93x + 44.79 | 102.03 | 0.20 |
| | 19 | BTD (10V-36V) | y = 1.08x + 63.64 | 102.18 | 0.17 |
| | 20 | BTD (10V-89V) | y = -0.92x + 116.53 | 102.17 | 0.18 |
| | 21 | BTD (18H-23H) | y = 3.18x + 77.95 | 102.19 | 0.17 |
| | 22 | BTD (18H-36H) | y = 0.18x + 83 | 102.23 | 0.16 |
| | 23 | BTD (18H-89H) | y = -1.4x + 122.75 | 102.09 | 0.20 |
| | 24 | BTD (18H-23V) | y = -3.92x + 75.13 | 102.26 | 0.17 |
| | 25 | BTD (18H-36V) | y = -0.51x + 90.15 | 102.24 | 0.16 |
| | 26 | BTD (18V-23H) | y = 6.1x + 32.36 | 102.08 | 0.20 |
| | 27 | BTD (18V-36H) | y = 0.86x + 68.45 | 102.21 | 0.17 |
| | 28 | BTD (18V-89H) | y = -1.14x + 122.94 | 102.13 | 0.19 |
| | 29 | BTD (18V-23V) | y = 6.35x + 61.65 | 102.13 | 0.18 |
| | 30 | BTD (18V-36V) | y = 0.43x + 78.64 | 102.22 | 0.16 |
| | 31 | BTD (18V-89V) | y = -1.4x + 126.53 | 102.10 | 0.20 |
| | 32 | BTD (23H-36H) | y = -0.09x + 86.33 | 102.23 | 0.16 |
| | 33 | BTD (23H-89H) | y = -1.57x + 123.73 | 102.06 | 0.20 |
| | 34 | BTD (23H-36V) | y = -1.16x + 93.49 | 102.23 | 0.17 |
| | 35 | BTD (23V-36V) | y = 0.68x + 74.51 | 102.22 | 0.16 |
| | 36 | BTD (23V-89H) | y = -1.36x + 125.26 | 102.09 | 0.20 |
| | 37 | BTD (23V-36V) | y = -0.07x + 86.26 | 102.23 | 0.16 |
| | 38 | BTD (23V-89V) | y = -1.62x + 126.92 | 102.06 | 0.20 |
| | 39 | BTD (36H-89H) | y = -2.10x + 113.67 | 102.01 | 0.20 |
| | 40 | BTD (36V-89H) | y = -1.91x + 118.52 | 102.03 | 0.19 |

Note: y = SD (cm)

**Reviewer comment 5**: The paper needs to describe how it deals with missing data in any of the many input datasets, if there is any, and if there is not, to clearly state that.

**Author Response:** In this study, there is no missing values in the input datasets and same has been mentioned in the section 3.1 i.e., data pre-processing section of the revised manuscript also.

*"There are no missing values for AMSR2 TB, SRTM elevation, SCD observations for the in-situ stations over WH region"*

**Reviewer comment 6**: The paper needs to explain the meaning of x_1 to x_13 in table 5 (see also line 371-372), without which the equations cannot be re-used by others.

**Author Response:** In the revised manuscript, meaning of $x_1$ to $x_{13}$ has been added. In Table 5 notes the details are updated as below.

*" $x_1$ to $x_5$ are latitude, elevation, slope, ruggedness, and SCD, respectively; $x_6$ to $x_{13}$ are the BTD of 10H18H, 10H23H, 18H89V, 36H89V, 36V89V, 23H89V, 10V89V, 10V23H, respectively; V is the vertical polarization, and H is the horizontal polarization; and 10, 18, 23, 36 and 89 is the frequency in GHz of the corresponding BT channels"*

**Reviewer comment 7**: Given the poor quality of the official AMSR2 snow product in the case study in this paper, the detailed analysis and comparison in table 7 is of very little interest to the community. It has already been established in Figure 4 and Table 6 that the AMSR2 product is not adequate in this example. Table 7 and associated text should ideally be removed.

**Author Response:** The authors partially agree with the opinion expressed by the reviewer. It is true that the poor performance of AMSR 2 is already established in the previous sections. However, the mean SD is diverse for same elevation and SCD classes in different WH zones. Further, given the topographic setting, the climatic and snow conditions are different in the same elevation range for different WH zones. Other than comparing both products, the section attempts to provide these heterogeneities and the variation in the model performance under these conditions. Hence, authors opine that the table can be of interest to some of the readers looking at how the model performance vary under these conditions. Therefore, authors suggest the reviewer to consider the inclusion of table into the manuscript as it is.

**Reviewer comment 8**: Line 441 the snow depth classes are supposed to be grouped according to in-situ measurements, but since this analysis is done over the whole Western Himalaya, there cannot be in-situ measurement everywhere, so it is not clear what is going on.

**Author Response:** In this study, SD classes are grouped according the in-situ measurement collected in three different zones of Himalaya (i.e., LHZ, MHZ, and UHZ). The same information is present in the L441 of the preprint and is rephrased for improving the clarity. Figure 4 shows the spatial distribution of mean SD of stations. In the figure it can also be seen that in-situ stations are distributed over all three WH zones.

*"In each WH zone, the AMSR2 SD products and multifactor SD model estimates are grouped into five SD classes, i.e., 0-25 cm, 25-50 cm, 50-75 cm, 75-100 cm, and >100 cm based on in-situ SD observations during 2017 -18 to 2018 -19."*

**Reviewer comment 9**: Line 321 "a total of 72 parameters" - as with the 32 BTDs that were rejected, it is as important to know which parameters did not correlate well with snow depth, as much as those that did. Hence, in case there are any other rejected parameters that have not been described in the text, it also needs to be clear what these were, at least in summary form.

**Author Response:** The authors regret the typographical mistake in the L321. It should be 57 parameters. The same has been used elsewhere in the manuscript. The authors agree with the reviewer's suggestion. This has been already addressed. Kindly refer to the response given to the comment 4.

**Reviewer comment 10**: The MODIS snow cover days predictor is not clearly enough defined. I still am not sure if it is the days per year (making this a fixed map) or if it is the presence of snow cover on that specific day the snow depth retrieval is made. The text probably needs to be clearer here (see also point 6)

**Author Response:** MODIS SCD indicates how many days in a year (i.e., 364/365 days), a pixel covered with snow. In Himalaya, snow cover area (SCA)/snow cover pixels vary during different months of the year due to change in snowfall and snow ablation pattern. Least SCA has been observed during the month of August/September and maximum SCA was observed during the month of February/March. Therefore, SCDs map will also change every day, month and years. For every day in a year SCD map has been generated and in the developed model SCDs map is variable. Reviewer 2 also have mentioned some suggestions with regard to SCD. The following changes have been incorporated in the revised manuscript.

*"In WH, snow cover area (SCA)/snow cover pixels vary during different months of the year due to change in snowfall and snow ablation pattern. Least SCA has been observed during the month of August/September and maximum SCA was observed during the month of February/March. Snow cover duration (SCD) depicts the number of consecutive days snow cover is present for a given pixel. It provides information regarding the persistence of snowpack and is useful in improving PMW SD estimates (Singh et al., 2016; Wang et al., 2019; Dai et al., 2018). In this study, daily could-free MODIS snow cover product (i.e., M\*D10A1GL06)*

*generated for high-mountain Asia (Muhammad and Thapa, 2020) at 500 m spatial resolution (https://doi.org/10.1594/PANGAEA.918198) has been used to generate SCD product for the study area during the data period. Previously, Sharma et al. (2014) and Singh et al. (2018) have generated and evaluated the SCD maps for snow-covered Indian WH. These studies (Sharma et al., 2014, Singh et al., 2018) revealed a higher average monthly SCD (>80%) in high-altitude regions. These studies' results further emphasize a strong longitudinal and altitudinal dependence on SCD, snow cover accumulation and ablation in WH.  Therefore, SCD information can provide valuable insights to improve the SD model. Daily binary snow cover maps prepared from M\*D10A1GL06 are used to identify the snow cover presence for a given pixel. These binary snow cover maps are used for computing the SCD information for each day from October 1st of each year to September 30th of the following year during the study period."*

**Response to Minor issues:**

**1:** Linę 264: "Grody's decision tree" should more fairly be "Grody and Basist's decision tree".

**Author Response:**  As suggested, we have updated in the revised manuscript.

*"Grody and Basist's decision tree makes use of different filters (see Figure. 3) based on the values of TB observations to separate snow from non-snow pixels"*

**Reviewer comment 2:** In defining Eq. 1- 4, "i" needs defining

**Author Response:**   Agreed. 'i' represents the number of parameters in the model. Accordingly, the information is updated into the manuscript.

*"where, y is the ground observed SD values; $x_1$, $x_2$, ......., and $x_i$ are the screened parameters; $\alpha_0$, $\alpha_1$, $\alpha_2$, …, and $\alpha_i$ are the regression coefficients of the multiparameter models, and i represents the number of parameters."*

**Reviewer comment 3:** It seems incorrect to have two constant offset coefficients in equations 1, 2, and 4, since it is only possible to estimate one. One of alpha_0 and c_i should probably be removed. In any case, only one constant offset coefficient is seen in Table 5, so the equations do not appear to be consistent.

**Author Response:** The authors sincerely apologize for this mistake and totally agree with the observations of the reviewer.  $\alpha_0$ is now removed the equations (7) – (11) in the revised manuscript (previously numbered as (1)-(4)); however, $c$ represents the error term in the equations. In the revised manuscript, changes have been incorporated as follows.

*"*

$$y = \alpha_1 x_1 + \alpha_2 x_2 + \cdots + \alpha_i x_i + c \qquad\qquad (1)$$

$$y = \alpha_1 Inx_1 + \alpha_2 Inx_2 + \cdots + \alpha_i Inx_i + c \qquad (2)$$

$$y = cx_1{}^{\alpha_1} x_2{}^{\alpha_2} \ldots\ldots\ldots\ldots x_i{}^{\alpha_i} \qquad (3)$$

$$y = \alpha_1 \frac{1}{x_1} + \alpha_2 \frac{1}{x_2} + \cdots + \alpha_i \frac{1}{x_i} + c \qquad (4)$$

"

**Reviewer comment 4:** Line 362 could mention the likely reason for why descending and ascending passes have different results, namely as stated on line 527 that one local time is during daylight and hence more prone to melting snow.

**Author Response:** The authors are thankful to the reviewer for the valuable suggestion. In the revised manuscript changes have been incorporated as given bellow.

**"***The SD models built with TB observations from descending orbital passes have relatively higher correlation and lesser RMSE compared to those from ascending pass TB data when analyzed with in-situ SD. This is mainly because descending orbital passes occur in the morning time with no melting of snow; however, ascending orbital passes occur in the afternoon time with substantial melting of snow in the study area. Therefore, only descending pass TB observations are used in the study.*"**

**Reviewer comment 5:** Line 384-385 - consider using a table rather than loading the text with so many numerical results.

**Author Response:** As suggested by reviewer, the lines 383-385 containing the numerical results are removed, and shown as Table 6 in the revised manuscript.

"*The R, RMSE (in cm) metrics of power, linear, logarithmic, and reciprocal models in different WH zones are shown in Table 6.*

*Table 6. Comparative analysis of multifactor SD models during 2017-2019 for WH zones*

| | Western Himalayan Zones | | | | | |
|---|---|---|---|---|---|---|
| | *Lower Himalayan* | | *Middle Himalayan* | | *Upper Himalayan* | |
| *Models* | *R* | *RMSE* | *R* | *RMSE* | *R* | *RMSE* |
| *Power* | 0.65 | 22.7 | 0.76 | 19.2 | 0.89 | 22.6 |
| *Linear* | 0.64 | 29 | 0.68 | 22.8 | 0.75 | 33.5 |
| *Logarithmic* | 0.38 | 52 | 0.14 | 41 | 0.73 | 36.9 |
| *Reciprocal* | 0.09 | 121.3 | 0.47 | 26.7 | 0.61 | 43.2 |

"